# Spatially resolved proteomics via tissue expansion

Lu Li[1,2,3,4,5,10], Cuiji Sun[1,2,4,10], Yaoting Sun [1,2,3,4,10], Zhen Dong [1,2,3,4,10], Runxin Wu[1,2,3,4,6], Xiaoting Sun[1,2,4], Hanbin Zhang [1,2,4], Wenhao Jiang[1,2,3,4], Yan Zhou[1,2,3,4], Xufeng Cen[7], Shang Cai[1,2], Hongguang Xia [7,8,9], Yi Zhu [1,2,3,4], Tiannan Guo [1,2,3,4] ✉ & Kiryl D. Piatkevich [1,2,4] ✉

Spatially resolved proteomics is an emerging approach for mapping proteome heterogeneity of biological samples, however, it remains technically challenging due to the complexity of the tissue microsampling techniques and mass spectrometry analysis of nanoscale specimen volumes. Here, we describe a spatially resolved proteomics method based on the combination of tissue expansion with mass spectrometry-based proteomics, which we call Expansion Proteomics (ProteomEx). ProteomEx enables quantitative profiling of the spatial variability of the proteome in mammalian tissues at ~160 μm lateral resolution, equivalent to the tissue volume of 0.61 nL, using manual microsampling without the need for custom or special equipment. We validated and demonstrated the utility of ProteomEx for streamlined large-scale proteomics profiling of biological tissues including brain, liver, and breast cancer. We further applied ProteomEx for identifying proteins associated with Alzheimer's disease in a mouse model by comparative proteomic analysis of brain subregions.

Complex biological events usually involve interactions of multiple biomolecules such as DNAs, RNAs, proteins, and metabolites, which are arranged with nanometer precision over extended length scales in biological systems. The development of spatially resolved multimodal omics methodologies (i.e., genomics, transcriptomics, proteomics, and metabolomics) provides essential support for a systematic understanding of complex cellular and molecular events within intact tissues[1–4]. One approach proved to be valid for improving the spatial resolution of biomolecules mapping is based on the physical magnification of samples via hydrogel embedding[4–6]. Enhancement of spatial resolution is achieved by the physical separation of biomolecules attached covalently to polymer chains while preserving their relative positions in the expanded state[5]. By treating the sample with specific chemical anchors, it is possible to retain various biomolecules including proteins and nucleic acids, which can be further interrogated in situ in expanded samples[7,8]. For example, tissue expansion has been recently combined with untargeted in situ genome and transcriptome sequencing enabling spatial mapping of hundreds of sequence-determined DNAs and RNAs per individual cell in their endogenous context[4,6]. However, in-depth proteomic analysis with high spatial

[1]Research Center for Industries of the Future and School of Life Sciences, Westlake University, 600 Dunyu Road, Hangzhou, Zhejiang 310030, China. [2]Westlake Laboratory of Life Sciences and Biomedicine, 18 Shilongshan Road, Hangzhou 310024 Zhejiang, China. [3]Key Laboratory of Structural Biology of Zhejiang Province, Westlake University, 18 Shilongshan Road, Hangzhou 310024 Zhejiang, China. [4]Institute of Basic Medical Sciences, Westlake Institute for Advanced Study, 18 Shilongshan Road, Hangzhou 310024 Zhejiang, China. [5]College of Pharmaceutical Sciences, Zhejiang University, 866 Yuhangtang Road, Hangzhou 310024 Zhejiang, China. [6]Whiting School of Engineering, Department of Biomedical Engineering, Johns Hopkins University, Baltimore, MD 21218, USA. [7]Department of Biochemistry & Molecular Medical Center, Zhejiang University School of Medicine, Hangzhou 310058, China. [8]Research Center for Clinical Pharmacy & Key Laboratory for Drug Evaluation and Clinical Research of Zhejiang Province, The First Affiliated Hospital, Zhejiang University School of Medicine, Hangzhou 310003, China. [9]Zhejiang Laboratory for Systems & Precision Medicine, Zhejiang University Medical Center, 1369 West Wenyi Road, Hangzhou 311121, China. [10]These authors contributed equally: Lu Li, Cuiji Sun, Yaoting Sun, Zhen Dong. ✉e-mail: guotiannan@westlake.edu.cn; kiryl.piatkevich@westlake.edu.cn

resolution using tissue expansion has not been fully demonstrated in biological samples[9]. Recent advances in mass spectrometry (MS) based proteomics have enabled systematic analysis of protein expression in various biological samples and preparations[10,11]. In particular, data-independent acquisition (DIA) MS offers superior depth and reproducibility of proteomic analysis over data-dependent acquisition (DDA)[12,13]. Improvements in MS sensitivity have enabled high spatial resolution proteomics of tissue samples by coupling MS-based analysis with different microsampling approaches involving either laser capture microdissection (LCM)[14,15], matrix-assisted laser desorption/ionization (MALDI)[16], or in situ microdigestion combined with surface extraction[17]. Several techniques based on these approaches can achieve up to ~50 μm of lateral resolution on thin (10's of μm) tissue sections[18–20], however, they require special equipment not accessible to most labs.

Here we present an accessible and easy-to-use approach for manual tissue microdissection coupled with bottom-up MS-based proteomic analysis to assess the spatial variability of the proteome in tissue at 100's of μm lateral resolution and sub-nanoliter volume precision. The microdissection is facilitated by physical magnification and staining of tissue via embedding it into a swellable hydrogel, similarly to the concept used in expansion microscopy (ExM) to improve imaging resolution. We demonstrated that by reversible anchoring proteins into hydrogel polymer network within biological tissue, the specimen can be isotropically magnified on average by 6.1-fold in linear dimension corresponding to 227-fold in volume and visualized by the naked eye, allowing precise manual microdissection of regions of interest based on fine anatomical/histological features or organ (sub) regions. We also developed subsequent peptide extraction procedures from the excised pieces of tissue-hydrogel composite for in-depth MS-based proteomic analysis with DIA-MS. Because of the utility of the developed procedures, we termed the process of tissue expansion and microdissection followed by peptides retrieval and MS-based analysis as expansion proteomics or ProteomEx for short. We benchmarked ProteomEx against traditional methods for MS analysis of tissue and demonstrated its utility for mapping quantitative proteome profiles on the morphology of mammalian tissues including brain, liver, breast cancer. Finally, we utilized ProteomEx for the identification of dysregulated proteins in the specific brain subregions in mice with Alzheimer's disease (AD).

## Results

### ProteomEx workflow development and optimization

To perform spatially resolved proteomics of biological tissue, we employed modified protein-retention ExM[5] followed by expanded sample microdissection, in-gel digestion, and MS-based proteomic analysis of the recovered peptides. In order to combine tissue expansion with liquid chromatography (LC)-MS/MS analysis, we developed and optimized key steps of tissue sample expansion, manual microdissection, and peptide extraction including i) an optimized hydrogel with enhanced expansion factor and mechanical stability; ii) reversible protein anchoring to polymer network; iii) isotropic expansion of sample; iv) sample staining; v) sample microdissection; vi) in-gel digestion and peptide extraction (see Supplementary Note 1, Supplementary Tables 1, 2, 3, and Supplementary Figs. 1, 2 for the details of the protocol development and optimization). As a result, we established a method for enhanced spatially resolved MS-based proteomic analysis of tissues via physical sample magnification, which we called ProteomEx. The ProteomEx workflow includes several sample treatment and interrogation steps as illustrated in Fig. 1A, B. First, chemically fixed tissue section, which can be immunostained if needed, is treated with N-succinimidyl acrylate to install acryloyl group on proteins primary amine groups thus enabling their reversible anchoring into polymer mesh via amide group. The treated sample is then embedded into the hydrogel, which was optimized for increased

expansion factor (5.5-8-fold in linear dimension, which corresponds to 166-512-fold expansion in volume, to achieve higher spatial precision during manual dissection) and enhanced mechanical properties appropriate for handling and manual sampling fully expanded samples (to avoid sample cracking and breaking). Then the tissue-hydrogel composite is subjected to a detergent-based treatment to render it mechanically homogeneous, which allows for both isotropic expansion of the embedded tissue and protein retention in the expanded state. Next, to facilitate tissue imaging and manual microdissection, the sample is stained with the colorimetric dye, Coomassie blue, enabling efficient visualization of the morphological details of expanded brain tissue by the naked eye and simple imaging setups (Fig. 1C–E). Then expanded sample can be imaged (to map proteome profile onto the tissue morphology) and manually dissected with ~100's micron precision to excise individual regions of interest or to microsample the tissue into the small voxels. Optimized in-gel digestion with alkaline buffer and trypsin is used to extract the peptides from excised tissue-hydrogel sections followed by LC-MS/MS analysis and data processing.

### ProteomEx validation and characterization

Since ProteomEx involves chemical treatment and expansion of tissue, it is important to quantify the efficiency of peptide extraction and validate the qualitative and quantitative reproducibility and sensitivity achievable with this method. First, we decided to measure peptide yield and missed cleavage rate as the major parameters of sample preparation quality control. For benchmarking, we used the well-established in-solution digestion and the pressure cycling technology (PCT)-assisted tissue digestion methods, which are widely used for tissue treatment in MS-based proteomics analysis[21]. While we were finalizing this study, Drelich et al. published a conceptually similar method[9], which we also used for side-by-side comparison with ProteomEx. Since the described method is based on the original protein-retention ExM protocol reported by Tillberg et al.[5], we refer to it as proExM-MS for convenience. It should be noted that we introduced several modifications to the peptide recovery procedure originally described by Drelich et al., which improved peptide yield and protein identifications (Supplementary Note 2 and Supplementary Fig. 3). Processing of the adjacent mouse brain tissue sections using the selected methods revealed that peptides extraction yield for ProteomEx was about 1.4–1.7-fold higher than that for in-solution digestion, PCT, and proExM-MS, specifically $72.54 \pm 6.17$ μg peptides/mg tissue (mean ± standard deviation (SD), throughout unless otherwise indicated for ProteomEx vs. $44.59 \pm 18.54$ μg, $43.58 \pm 11.59$ μg, and $50.64 \pm 9.33$ μg for in-solution, PCT, and proExM-MS, respectively; Fig. 2A). Furthermore, ProteomEx was characterized by a lower number of missed cleavages ($20.04 \pm 1.28\%$) compared to $27.96 \pm 1.38\%$ and $24.07 \pm 1.22\%$ for in-solution and PCT protocols, respectively, although similar to that for proExM-MS ($21.17 \pm 5.17\%$; Fig. 2B). The higher efficiency of peptide digestion and extraction achieved with ProteomEx and proExM-MS can be probably explained by molecular decrowding in the expanded state providing better access for enzyme molecules to the proteolytic sites. These results indicated that the tissue expansion protocol used in ProteomEx provided higher efficiency of peptide extraction compared to in-solution and PCT-assisted sample preparation methods for tissue samples as well as the conceptually similar proExM-MS.

Next, the extracted peptides were analyzed using a timsTOF Pro mass spectrometer in data-dependent acquisition (DDA) mode. By processing ~200 ng of peptides from each sample, we identified 37,173, 34,304, 20,413, and 30,630 peptides corresponding to 4199, 4278, 3181, and 3818 individual proteins on average per sample prepared using in-solution, PCT, proExM-MS, and ProteomEx methods, respectively (Fig. 2C, D; Supplementary Fig. 4). Variability of peptide and protein numbers were lower for ProteomEx, proExM-MS, and PCT than

that for the in-solution method, suggesting a higher degree of reproducibility of these methods. The number of proteins identified with ProteomEx was lower by about 400 proteins than that identified with in-solution and PCT methods, though higher by 637 proteins than that for proExM-MS.

The diversity of all identified proteins with the four methods had an overlap of 56.6%, and each method uniquely identified less than 7.0% of the total number of proteins (Fig. 2E). The overlap for the identified proteins was reasonable (>50-60% is the typical overlap for DDA mode for data analysis) and high enough to indicate that ProteomEx performance was comparable to the methods used for sample processing in MS-based proteomics. Furthermore, all four methods exhibited similar distribution of identified proteins by biomarkers, subcellular localization, and biological function (Fig. 2F,

Supplementary Fig. 4). To verify how chemical treatment during the ProteomEx procedure can modify amino acids, we searched for the post-translationally and chemically modified peptides obtained with four used sample preparation methods. The data analysis revealed that only a very minor fraction (<0.5%) of peptides extracted with ProteomEx had chemical modifications associated with the *N*-succinimidyl acrylate (NSA) anchoring while recovery of naturally occurring post-translational modifications was similar to other methods (Supplementary Note 3 and Supplementary Fig. 5). Overall, the above results demonstrated that ProteomEx could acquire a high-quality proteome and was comparable with the other available methods in terms of the number and type of protein identifications, post-translational modifications recovery, which met the proteomic analysis needs.

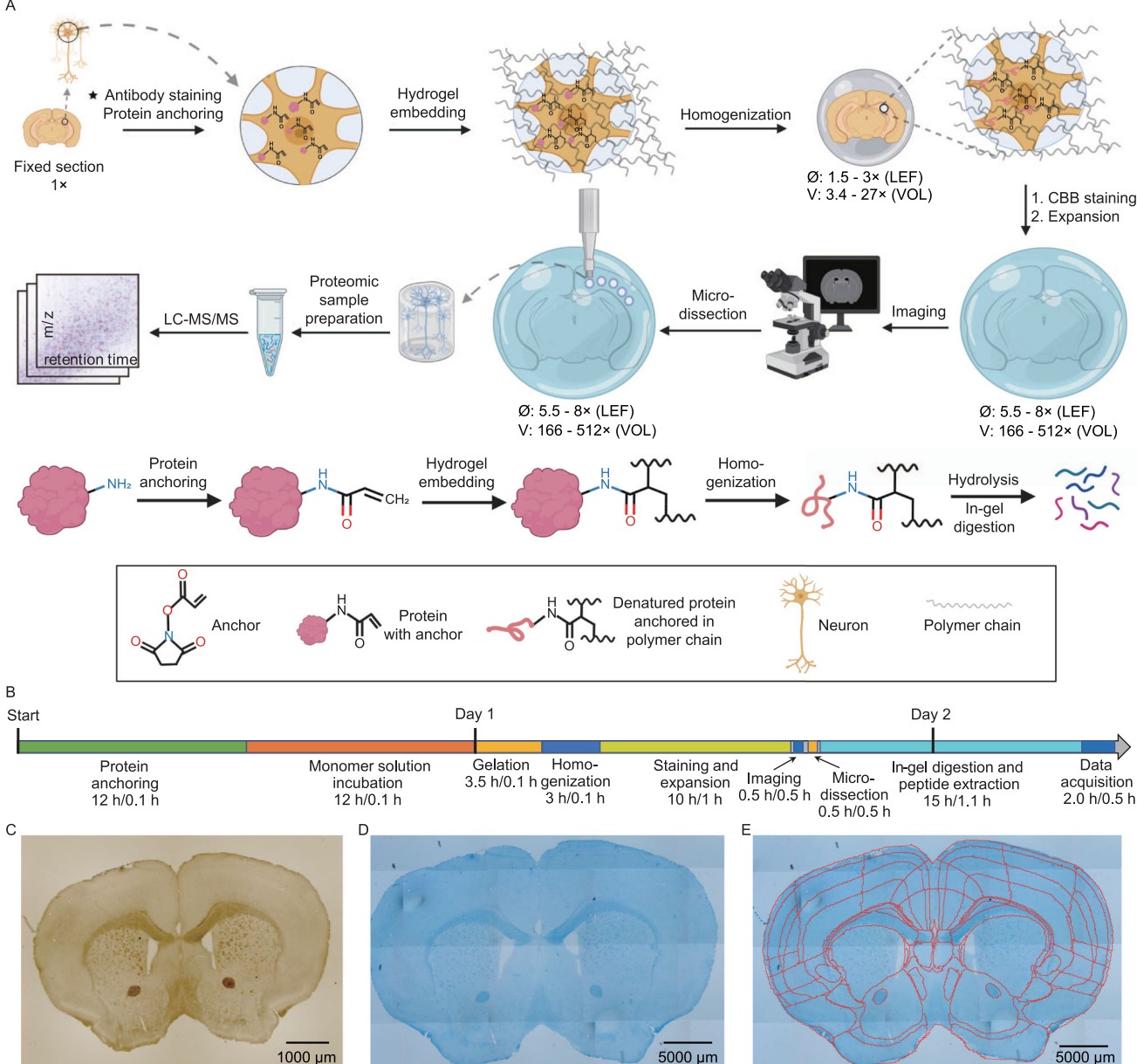

**Fig. 1 | ProteomEx workflow. A** Chemically fixed tissue samples, which can be immunostained beforehand, are treated with the chemical anchor, embedded into the hydrogel, and mechanically homogenized by mild denaturation. The Coomassie brilliant blue (CBB) hydrogel embedded samples are expanded and imaged. After imaging, expanded samples are microdissected and excised pieces of the tissue-hydrogel composite are processed by in-gel digestion to recover peptides for LC-MS/MS analysis (LEF linear expansion factor, VOL volumetric expansion factor). Created with Biorender.com. **B** Timeline of ProteomEx indicating total duration and hands-on time of each step (total duration/hands-on time). **C** Representative brightfield images of mouse brain tissue section before expansion and (**D**) after Coomassie staining and expansion (**E**) showing automatically detected and annotated brain regions (LEF = 5.5-fold; $n$ = 20 brain slices from 16 mice).

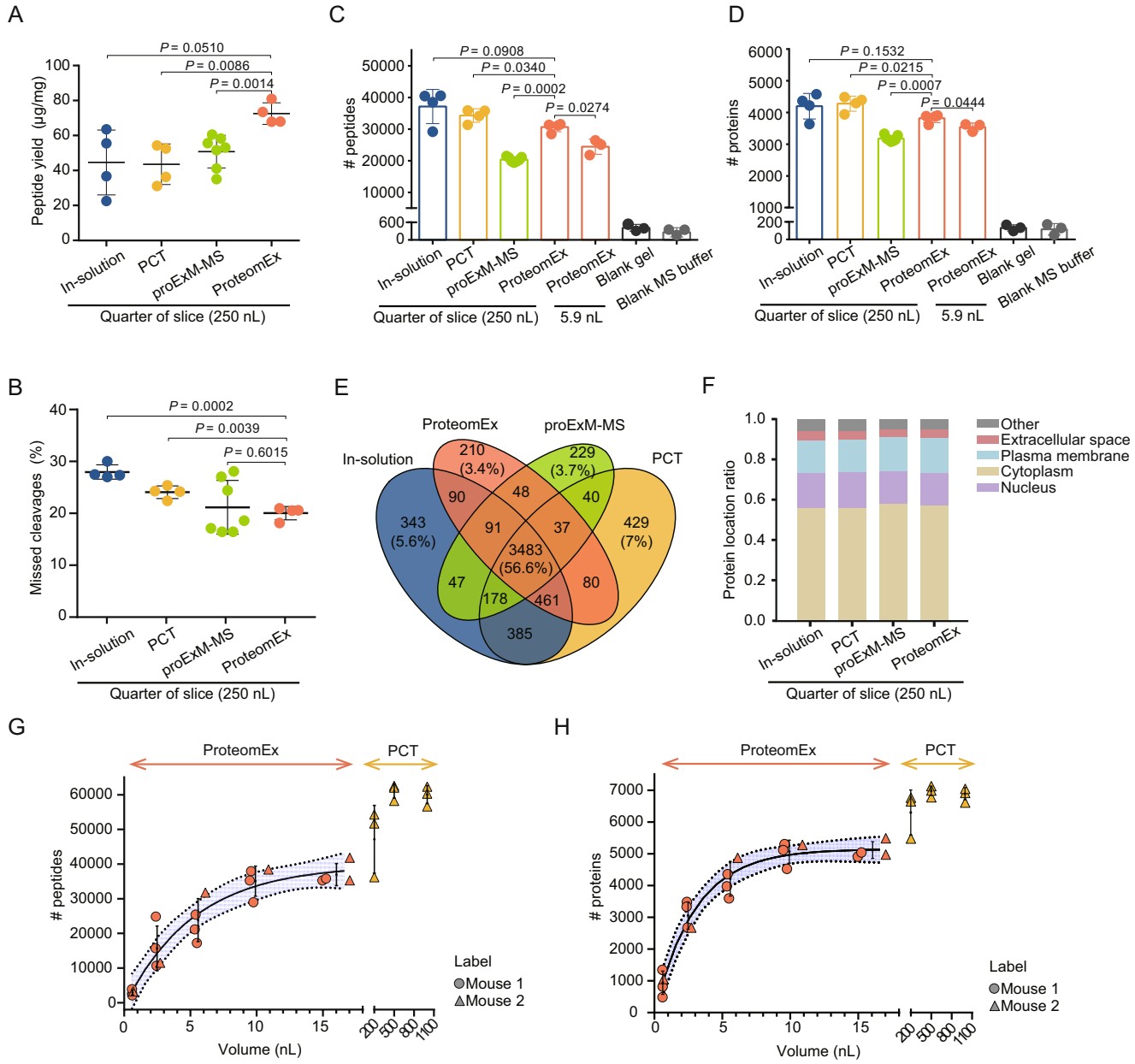

**Fig. 2 | ProteomEx benchmarking. A** The peptide yields of the in-solution, PCT, proExM-MS, and ProteomEx methods applied to the mouse brain tissue ($n$ = 4, 4, 7, 4 biologically independent samples from one, one, two, and one brain slices, respectively, the same samples were used to acquire data shown in panels **A**–**F**; dot, individual data point, bar, mean, whiskers, standard deviation (SD), throughout Fig. 2; *P*-values are estimated by Welch's *t*-test (two-sided). Data are presented as mean values ± SD.). **B** Missed cleavages of the identified peptides prepared using in-solution, PCT, proExM-MS, and ProteomEx methods. Data are presented as mean values ± SD. Number of peptide (**C**) and protein (**D**) identifications in seven sample groups ($n$ = 3 and 3 punches from one slice from one mouse for ProteomEx (5.9 nL) and blank hydrogel, $n$ = 3 independent injections

for MS buffer; analyzed by DDA). Data are presented as mean values ± SD. **E** Venn diagram of identified proteins for the bulk samples shown in **D**. **F** The subcellular locations of the identified proteins for the samples shown in **E**. Number of peptide (**G**) and protein (**H**) identifications for different tissue volumes processed by ProteomEx and PCT ($n$ = 4 punches per group from 2 mice for ProteomEx, LEF = 6.3, 6.2, 6.3, 5.9; $n$ = 3 tissue dissections per group from 1 mouse for PCT; dot and triangle, individual data point; center of error bar ends, mean; whiskers, SD; solid line, four-parameter logistic fit, dashed line indicates 95% confidence interval border, shaded area represents 95% confidence interval; analyzed by PulseDIA; dot, individual data point; bar, mean; whiskers, SD. Source data are provided as a Source Data file.

Staining of the expanded tissue with the colorimetric dye facilitated visualization of fine morphological features with the naked eye and precise targeting of a region of interest by manual microdissection (down to ~100 μm of real size). To validate this capability for MS analysis, we used a biopsy punch, which provided highly reproducible microsampling, to excise 3 mm-diameter tissue-hydrogel composite pieces (corresponding to ~500 μm in diameter or 5.9 nL tissue volume before expansion, here and throughout the size corresponds to that before expansion) and adjacent blank hydrogel pieces (used as a

control of possible peptide diffusion outside of tissue) from the same expanded mouse brain tissue slice and analyzed as described above using pure MS buffer as a negative control. We identified 24,437/3541 peptides/proteins on average per punched sample, which was much higher than that for the blank hydrogel (416/132 peptides/proteins) and MS buffer (260/116 peptides/proteins corresponding to carry-over level; Fig. 2C, D), and similar protein distribution by subcellular localization and biological function as observed for the bulk sample analysis (Supplementary Fig. 4). To further benchmark ProteomEx, we

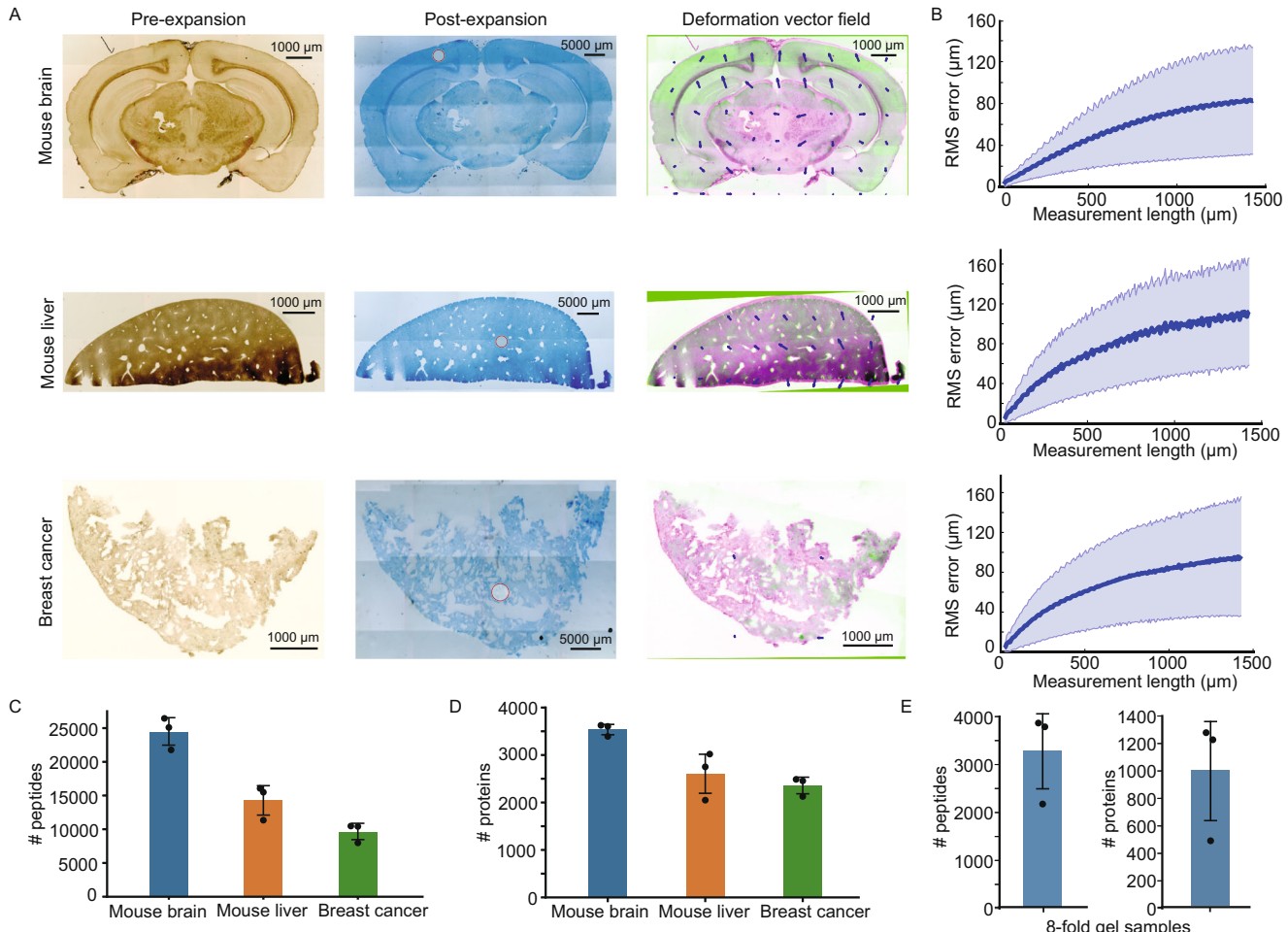

**Fig. 3 | Validation of ProteomEx in different tissue types. A** Representative bright-field images of different tissue slices pre-expansion (left column) and post-expansion (middle column; Coomassie-stained) and overlay (right column) of pre-expansion image (green pseudo-color) and registered post-expansion image (magenta pseudo-color). Arrows represent the deformation vector field ($n = 3, 3,$ and 3 tissue slices from one mouse each; in the images of the expanded sample white circles represent the dissected area for peptide analysis shown in **C. B** The root-mean-square (RMS) measurement length error for pre- versus post-expansion brain slice images for the experiments shown in **A** ($n = 3, 3,$ and 3 tissue slices from one mouse each; average LEF = 5.8, 6.5, and 6.2 for brain, liver, and tumor respectively; blue line, mean; shaded area, standard deviation (SD)). Number of peptide (**C**) and protein (**D**) identifications ($n = 3, 3,$ and 3 punches for each tissue from one mouse each; 5.6 nL tissues were punched out and analyzed by DDA; dot, individual data point, bar, mean, whiskers, SD). Data are presented as mean values ± SD. **E** Number of peptide and protein identifications for 0.37 nL brain tissue analyzed in PulseDIA mode ($n = 3$ punches from one slice; dot, individual data point, bar, mean, whiskers, SD). Data are presented as mean values ± SD. Source data are provided as a Source Data file.

evaluated the technical reproducibility of protein quantitation by calculating the Pearson correlation between each pair of macro- (250 nL) and microsamples (2.75–17.19 nL) and coefficient of variation (CV) values for each size group. To minimize the biological variability of microsample preparations, we used mouse liver tissue, which is more homogenous than brain tissue. The ProtemoEx method showed comparable reproducibility of protein quantification for macro-samples and similar or slightly lower reproducibility for microsample preparations compared to other tested methods (Supplementary Figs. 6, 7). Thus, peptides can be efficiently extracted from the small pieces of the expanded tissue with sufficient technical reproducibility and neglectable diffusion into the blank hydrogel around the tissue. Additionally, compared with whole-brain slices processed by ProteomEx, 3 mm gels identified a slightly lower number of peptides while a comparable number of proteins.

Next, we explored the volume-dependent limit of tissue microsampling using ProteomEx approach. Processing the microdissected mouse brain coronal sections with actual volume of about 0.6 nL (0.2 µL of tissue-hydrogel composite), 2.4 nL (0.9 µL), 5.4 nL (2.1 µL), 9.6 nL (3.8 µL), and 15.0 nL (5.9 µL; corresponding to lateral resolution of about 160, 320, 480, 640, 800 µm on 30 µm tissue section), we identified 2987, 15,705, 23,898, 35,160, and 37,071 peptides corresponding to 928, 3044, 4203, 5058, and 5105 unique proteins, respectively, on average per size group (Fig. 2G, H). As expected, the numbers of identified peptides and proteins increased with tissue volume reaching a plateau at around 5.0 nL tissue size (or 480 µm in diameter). PCT-assisted sample preparation, as a representative method for processing small samples, enables effective analysis of tissues volume in the range of 0.2–1 µL, which is about three orders of magnitude higher than that for ProteomEx. Thus, ProteomEx provides an alternative strategy for tissue proteomic analysis for sub-nanoliter volume sample preparation at ~100's-µm lateral resolution.

To assess the applicability of ProteomEx to various mammalian tissues, we performed ProteomEx on four different mouse tissue types including brain, liver, breast cancer, and lungs (Fig. 3A and Supplementary Fig. 8). Since ProteomEx utilized hydrogel composition and optimized homogenization treatment not applied before for tissue expansion, we first quantified the isotropy of hydrogel-based tissue expansion using a non-rigid registration as done previously for the original protein-retention ExM method[5]. The isotropic expansion is

essential for precise mapping of spatial proteome distribution onto pre-expanded tissue morphology. We calculated the root-mean-square (RMS) length measurement error of feature measurements after tissue expansion over length scales up to 1500 μm and found that RMS errors were ~8%, ~10%, ~8%, and ~10% of the measurement distance for brain, liver, breast cancer, and lungs tissue samples, respectively (Fig. 3B and Supplementary Fig. 8). Next, we processed ~5 nL volume of each tissue type using DDA-MS and identified 24,436/3540, 14,298/2606, 9623/2356 peptides/proteins for brain, liver, and breast cancer samples, respectively (Fig. 3C, D). To explore the possibility to correlate proteomic profile with cellular and subcellular features visualized via immunohistochemistry and small dye staining, we stained the AD mouse brain slice with DAPI and Aβ antibodies, imaged and processed it using ProteomEx. For the 2.52 nL volume of the immunostained tissue, we identified ~7000 peptides corresponding to ~2000 proteins for three replicates demonstrating the compatibility of ProteomEx with immunohistochemistry (Supplementary Fig. 9). ProteomEx can be readily applied to different mammalian tissue types and is compatible with antibody-stained samples.

To explore the limits of lateral spatial resolution and tissue volume for ProteomEx, we expanded the brain tissue section by 8-fold in linear dimension (512-fold in volume) and punched out 1-mm diameter tissue-hydrogel composite corresponding to the pre-expansion diameter of 125 μm for proteomic analysis (Supplementary Fig. 10). The pre-expansion volume of punched tissues was 0.37 nL, equivalent to approximately 160 cells (calculated using BNID 100434). On average, we identified and quantified more than 3000 peptides and ~1000 proteins per sample analyzed in PulseDIA mode[22] (Fig. 3E).

## Proteomic profiling of normal and pathogenic brain tissue with subregion precision

Having demonstrated that ProteomEx enabled straightforward proteomic profiling of sub-nanoliter volume of tissue at hundreds-micron lateral resolution, we applied this technique to characterize proteomic heterogeneity of mouse brain with and without AD. We used the APP/PS1 mouse model of AD and wild-type (WT) mice in two age groups with a primary focus on the hippocampus as one of the most predominant regions of AD pathology[23] (Fig. 4A). The coomassie-stained expanded brain tissue had distinct anatomical landmarks allowing to pinpoint subregion of interest by the naked eye without a need for any imaging system assistance. From each mouse brain, we dissected multiple subregions (~330 μm diameter before expansion) from the primary visual area of the cortex (V1), hippocampal field CA1 (CA1), hippocampal field CA3 (CA3), dentate gyrus (DG), and medial geniculate complex (MGC) using ProteomEx with a LEF of ~6 (Fig. 4A). Microsampling was done manually using a 2-mm biopsy punch after tissue expansion, which enabled selective and reproducible microdissection of subregions. We chose this sampling size as it was sufficient to selectively dissect brain subregions and, based on our previous results, provided sufficient depth of proteome identification (Fig. 2G, H). To verify manual microsampling precision for each excised sample, the expanded tissues were imaged before and after dissection. The images were registered and annotated using the Allen Institute brain atlas. In total, we excised 144 hydrogel samples from the 12 mice and processed them in parallel using the optimized in-gel digestion protocol. Using PulseDIA–MS[22] on a timsTOF Pro mass spectrometer, 106,892 peptide precursors, 51,203 peptides from 6233 proteins were identified. After quality control analysis as described in the Methods and Supplementary Fig. 11, 6215 unique proteins quantified in 122 samples were subjected to downstream data analysis.

In the global view, all the 122 samples were well resolved by age (old/young) in the t-distributed stochastic neighbor embedding (t-SNE) plot (Fig. 4B), but different genotypes (AD/WT) can only be identified in old mice. The brain tissues were well distinguished by genotype for the old age groups, but the young age groups exhibited

no significant difference rather than for two differentially expressed proteins (DEPs), APP and PSEN1, as expected based on the genotype of the used AD mice (Fig. 4C). Since the proteomic changes were more pronounced in the old groups, we focused on the old groups for further bioinformatics analysis.

In the old groups, the proteomic alterations in the AD mouse brain varied among different regions or subregions. At the region level, there were 73 DEPs in the hippocampus while only six in the cortex and one in the MGC (Fig. 4C). These findings were consistent with previous studies reporting that the main lesions of AD occur in the hippocampus[24]. The syntaxin binding protein 2 (STXBP2), Apolipoprotein E (APOE), and Clusterin (CLU) were overlapped DEPs in the cortex and hippocampus but not in MGC. APOE and CLU have been previously associated with AD progress[25–27] (Fig. 4D). In the case of STXBP2, only a few studies have found its expression in the brain[28,29] however, there is no literature data reporting its association with AD. Our study uncovers its potential involvement in AD.

The subregional changes of the proteome in the hippocampus were also significant. CA1 had the most substantial alteration with 198 DEPs, followed by CA3 with 19 DEPs, while DG had no obvious changes (Fig. 4C), suggesting that the changes of AD lesion were not spreading in the entire hippocampus, but in the subregion of hippocampus, CA1 and CA3. APOE, STXBP2 as well as Proline-rich protein 7 (PRR7) and vesicle-associated membrane protein 1 (VAMP1) were changed in CA1 and CA3 but not in the DG subregion (Fig. 4D). More importantly, STXBP2 and APOE were the common DEPs in multiple regions, which proves the pivotal role of STXBP2 again (Fig. 4D). PRR7 and VAMP1 were only differentially expressed in CA1 and CA3 (Supplementary Fig. 11C). These results indicated that the ProteomEx workflow enables effective investigation of the pathological heterogeneity of AD.

We further showed the protein expression of STXBP2 for each punch in a spatial proteomic map (Fig. 4E and Supplementary Fig. 11D). STXBP2 was almost not expressed in all regions from the AD group but was highly expressed in the WT group. In the WT group, STXBP2 was highly expressed in cortex V1 and MGC, while a relatively lower expression level was detected in subregions of the hippocampus.

Due to the largest alteration in CA1 of the old AD mouse, next, we focused on the pathways and functions changes in CA1. 192 DEPs were enriched by the Ingenuity Pathway Analysis (IPA) and Metascape databases. The top ten enriched pathways and the participating proteins as well as the relationship between the pathways and proteins are shown in Fig. 4F. The most significant negative pathway was the signaling by Rho family GTPases, which has been shown to transduce extracellular signals to the actin cytoskeleton to modify axon outgrowth and growth cone motility. In addition, some proteins are involved in multiple signaling pathways simultaneously, for example, GNAI3, a transducer of G protein-coupled receptors (GPCRs). It participates in the RHOGDI signaling, i.e., signaling by Rho Family GTPases and the opioid signaling pathway. To further explore the biological process and protein-protein interaction, the top-three significantly enriched clusters are shown in Supplementary Fig. 11E by the MCODE algorithm. STXBP2 and VAMP1 as the overlapped proteins in the hippocampus played key roles in axon guidance, nervous system development, and developmental biology.

We next explore the spatial heterogeneity of the hippocampus by comparing the protein expression for three hippocampus subregions within the same group. The much lower differences in old AD groups (five DEPs) than in the other three groups (~40 DEPs; Supplementary Fig. 11F), which indicated spatial heterogeneity was partially weakened. We then calculated Pearson correlation coefficients between every two samples from the three regions and showed the results in an unsupervised hierarchical clustered heatmap. Most of the samples from the same subregions were clustered together regardless of genotype or age group (Fig. 4G). The samples from DG show high consistency on protein expression level, indicating small variation among the different

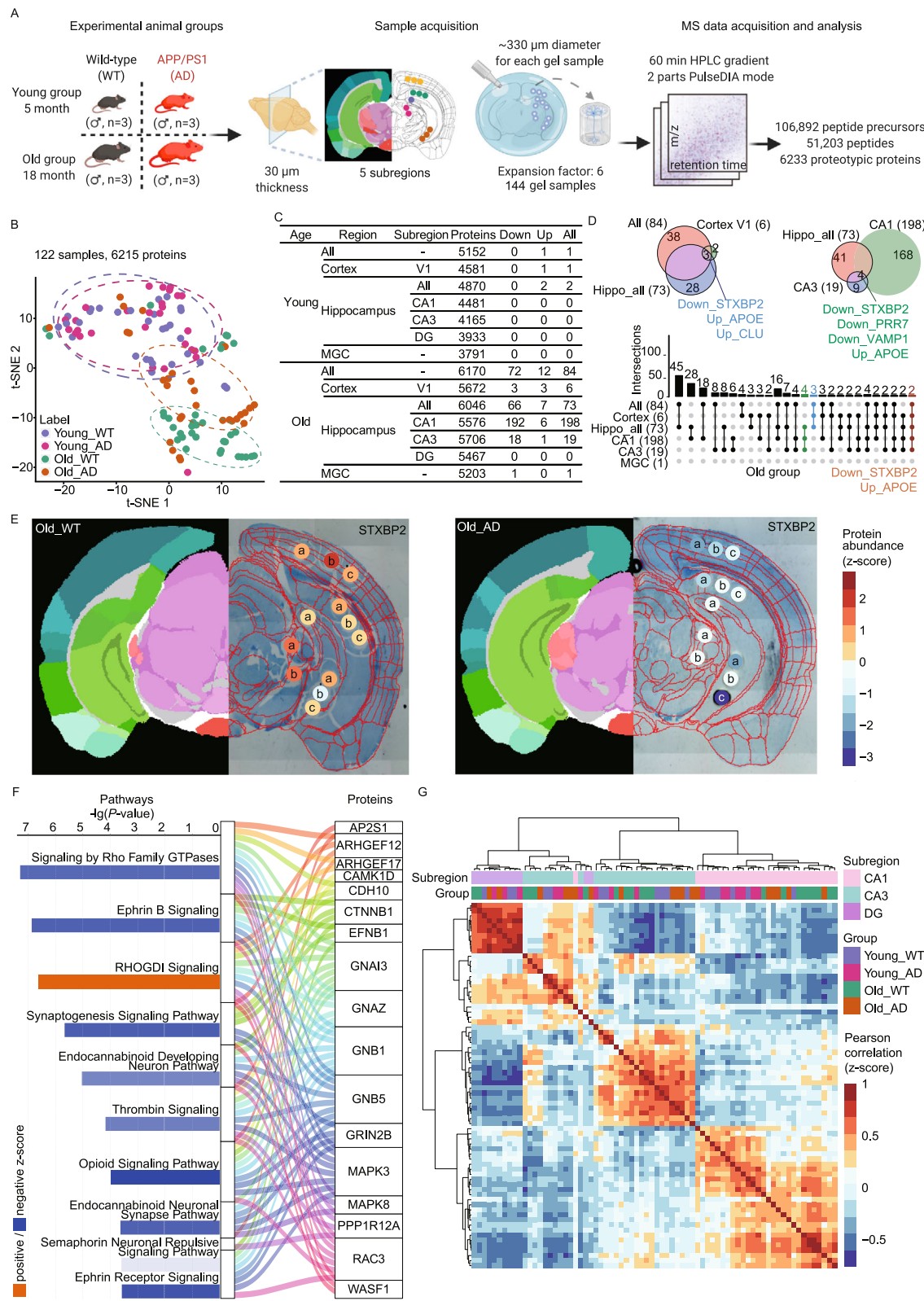

groups. In the case of the CA3 region, the analyzed samples can be further divided into two clusters based on protein expression, which needs to be validated in the future. In the cluster of CA1, samples from old mice were well separated from those from young mice. The above data suggest that ProteomEx can be used as a powerful tool to investigate spatial proteomics.

## Discussion

In the present study, we developed and validated a method for effective spatial proteomic profiling of biological tissues of sub-nanoliter volume at 100's of micron lateral resolution. The ProteomEx method combines specimen magnification via embedding in the swellable hydrogel with MS-based analysis of peptides extracted by optimized

**Fig. 4 | ProteomEx applied to AD mouse brains. A** Study design of proteomic analysis of the wild-type and AD mouse model representing (1) experimental animal groups (*n* = 3 mice per group), (2) sample acquisition, (3) MS data acquisition and analysis. Brain subregions selected namely, primary visual cortex (V1, *n* = 3 punches per slice per mouse), hippocampal field CA1 (CA1, *n* = 3 punches per slice per mouse), hippocampal field CA3 (CA3, *n* = 3 punches per slice from per mouse), dentate gyrus (DG, *n* = 1 punch per slice per mouse), and medial geniculate complex (MGC, *n* = 2 punches per slice per mouse). Created with Biorender.com. **B** t-SNE plot showing the sample clusters based on the prototype (*n* = 3 mice per group). **C** Number of differentially expressed proteins in mouse brain. **D** Venn and upset diagrams showing the DEP overlaps for selected regions. **E** Representative spatial proteomic maps of syntaxin binding protein 2 (STXBP2) in old WT (left) and old AD (right) brain slices (the left half shows the color annotation of mouse brain structures, the right half is a bright-field image of Coomassie-stained expanded brain tissue overlaid with punched locations, circles, and automatically registered atlas diagram of anatomical structures, red lines; LEFs were 5.9 ± 0.2 and 5.8 ± 0.2 for WT and AD, respectively; *n* = 6 and 6 slices from 6 WT and 6 AD mice, respectively). The color bar represents the z-score normalized protein abundance for the punches. The a/b/c in the punches represents biological replicates from the same brain region. **F** Pathway enrichment of 192 DEPs in CA1 and Sanky plot exhibiting the correlation between enriched pathways and proteins (*P*-values were calculated by the right-tailed Fisher's exact test based on the IPA database). **G** The hierarchical clustering heatmap showing the z-score scaled Pearson correlation coefficients between two samples labeled by subregions of hippocampus and mouse group. The Pearson correlation coefficients were estimated by the abundance expression of 101 proteins. Source data are provided as a Source Data file.

in-gel digestion from microdissected pieces of the tissue-hydrogel composite. The ProteomEx protocol, including tissue expansion, visualization, microdissection, and peptide extraction, is robust, cheap, easy to use, and can be readily deployed in a regular lab using commercially available reagents and common supplies. The optimized in-gel digestion procedure provides high efficiency of peptide recovery compatible with any MS-based proteomics workflow. The total duration of the protocol is about 58 h starting from the fixed tissue; however, the hands-on time is only 5 h and the longest steps, including tissue expansion and in-gel digestions, can be performed for multiple samples in parallel. Depth of protein profiling with ProteomEx was only ~10% lower than that achieved with the commonly used methods for bulk tissue preparation, although it did not result in substantial loss of important biological information including biomarkers and was characterized by an almost identical distribution of subcellular localization (Fig. 2F and Supplementary Fig. 4). Overall, ProtomEx enabled quantitative protein identification almost indistinguishable from the common sample processing methods.

We demonstrated that using manual microdissection we could efficiently and reproducibly achieve the lateral resolution of about 160 μm, which corresponds to ~262 cells or 0.61 nL tissue volume before expansion. This lateral resolution is comparable to that usually achieved with state-of-art techniques such as LCM[19,30,31] used in combination with bottom-up proteomics. From another hand, ProteomEx provides at least twice higher lateral resolution compared to alternative microsampling approaches based on liquid extraction surface analysis (LESA)[32–34]. We also demonstrated possibility to achieve ~125 μm later resolution (or ~1 mm of expanded tissue) by expanding sample by 8-fold in linear dimension (Fig. 3E and Supplementary Fig. 10), however, the major reason we did not exceed this resolution was difficulties in manual handling of small transparent gel samples, which were hard to see by the naked eye. Enabling the handling of submillimeter gel pieces, for example, by employing robotics or microfluidics, can further improve the spatial resolution of ProteomEx. It is also should be noted that distortion observed for the expanded samples on a macroscopic scale (>1000 μm) was about 2–4 times higher than that reported for the ExM-based method for super-resolution imaging on a microscopic scale (<100 μm)[5,35], and thus should be also considered when mapping protein profile on tissue morphology.

Complementary to microdissection, peptide recovery from trace samples is another challenge for spatially resolved proteomics. To date, nano-droplet processing in one-pot for trace samples (nanoPOTs)[18] is one of the most efficient methods for peptide recovery from nanoliter and subnanoliter volume samples. Compared to nanoPOTs technique, ProteomEx yielded a similar number of protein identifications for ~1 nL tissue volume although protein profiling depth for subnanoliter volumes was lower with ProteomEx (Supplementary Table 4, Supplementary Fig. 12). We also compared peptide and protein identifications for trace samples achieved with ProteomEx to that achieved with alternative peptide recovery methods including in-

column trypsin-digest system[36], immobilized enzyme reactors[19], and LESA[37] based on the literature data (Supplementary Table 4, Supplementary Fig. 12). Due to lower requirement for special equipment and infrastructure, ProteomEx is a more accessible technique for spatially resolved proteomics compared to the LCM- and LESA-based methods, which are on expensive and sophisticated hardware including LCM system, nanowell chips, and robotic microfluidic setups. However, it should be noted LCM enables higher spatial resolution than ProteomEx with the ability to cut arbitrary shapes. The practicality of ProteomEx allowed us to perform large-scale proteomic profiling of brain subregions from multiple brain samples and to identify differentially expressed proteins related to AD. As a proof-of-principle application, we were able to map proteome of the 144 hydrogel microsamples from the 12 mice in the course of 58 h starting from fixed brain slices. PulseDIA analysis of the peptides prepared by ProteomEx from these samples allowed us to identify multiple DEPs in the brain subregions.

While this manuscript was in preparation, Drelich et al. reported conceptually similar approach for spatially resolved proteomics[9], which we refer to as proExM-MS for short. Although both proExM-MS and ProteomEx share the same method for sample magnification based on proExM[5], they differ in several aspects crucial for their applicability. First, the used hydrogels are characterized by different expansion factors. The expansion factor of proExM-MS is only ~3-fold, which is 2.7-times lower than that achieved with ProteomEx. Correspondingly, ProteomEx provides the higher lateral resolution of microsampling achieving ~160 μm with manual dissection vs. ~328 μm for proExM-MS. The improved spatial resolution extends applicability of ProteomEx as it is more biologically relevant for analysis of heterogeneity of biological tissues. For example, ~100 μm resolution is more appropriate for analysis of functional tissue units (FTU)[38], e.g., renal glomeruli[39–41], colonic crypts[39], human lung blood vessels[36], which are typically limited by 100 μm in size. Furthermore, Piehowski *et al.* demonstrated that 100 μm resolution was sufficient to characterize region-specific bioactivity and unique tissue microenvironments within the mouse *Wnt5a*-null uterine tissue[30]. Second, protein anchoring is performed with different NHS-ester derivatives, which perhaps can explain difference in peptide yields. Previously we demonstrated that Acryloyl-X utilized as chemical anchor in proExM-MS has limited penetration depth in brain tissues treated at neutral pH and room temperature resulting in lower protein retention in expanded state[5]. Moreover, the N-succinimidyl acrylate chemical anchor used in ProteomEx is more chemically stable and significantly more affordable reagent compared to Acryloyl-X thus making ProteomEx more accessible and user-friendly technology. In addition, we extended applicability of ProteomEx to a variety of mammalian tissues and tissue staining methods including immunohistochemistry and small colorimetric dyes. Colorimetric staining is particularly crucial. All tissue samples upon expansion become transparent to a point when it is hard or almost impossible to visualize tissue outline and features by naked eye, compare for example 2.3-fold expanded tissue without staining with 6- and 8-fold expanded tissue post Coomassie staining

(Supplementary Figs. 3, 9, 10). Therefore, precise manual micro-dissection, when it is needed to pinpoint particular region based on morphological features, is impossible without staining. Furthermore, staining allows to confirm isotropic expansion and map proteome profile on tissue morphology. Compatibility with immunostaining and small fluorescence dyes, e.g., DAPI, opens up unprecedented capabilities of correlating super-resolution cellular and subcellular morphology with in-depth proteome analysis.

The ProteomEx technology enables a practical and effective alternative approach for spatially resolved MS-based proteomics analysis of fixed tissues of sub-nanoliter volume, which is otherwise only achievable using sophisticated equipment such as LCM. Since ProteomEx resembles protein-retention expansion microscopy, it can be combined with super-resolution microscopy of cellular structures and DNA and RNA fluorescence in situ hybridization enabling a spatially resolved multi-omics approach. Therefore, the ProteomEx is a potential technology for achieving high-resolution proteomics with deep sequence coverage.

## Methods

### Hydrogel screening

The monomers and crosslinkers were mixed in pure water at 23 °C in various combinations (see Supplementary Note 1 for details), supplemented with the corresponding initiator, and polymerized either in 1.5 ml Eppendorf tube or 3.5 cm MatTek dish (P35G-1.5-14-C) in humidified $N_2$ atmosphere using a vacuum oven (DZF-6000, Shanghai Sunrise Instrument) at respective temperature for 2–4 h. Chemicals with low water solubility were first dissolved in tetrahydrofuran at the appropriate concentration and diluted with water to the final concentration. The stock solution of VA-044, V-50, ammonium persulfate, and potassium persulfate (all from Signal Aldrich) was freshly prepared in water and used at final concentrations 0.2%, 0.6%, 0.2%, and 0.4%, respectively. After polymerization, the gels were removed from the gelation chamber and placed in excess volumes of doubly deionized water for 1–2 h to expand, with longer times for thicker hydrogels polymerized in Eppendorf tube. This step was repeated 3–5 times in fresh water, until the size of the expanding hydrogel sample plateaued. Stress–strain curves of expanded hydrogels were measured using the compression testing machine (CTM6050, Xie Qiang Instrument Manufacturing Co, China) equipped with a S9M/1 kg force sensor (HBM, Germany).

### Animal care and procedures

All animal maintenance and experimental procedures were conducted according to the Westlake University Animal care guidelines, and all animal studies were approved by the Institutional Animal Care and Use Committee (IACUC) of Westlake University, Hangzhou, China under animal protocol #19-044-KP.

Male transgenic APP/PS1 (on a C57BL/6J background) mice were provided by Prof. Hongguang Xia from Zhejiang University, female MMTV-PyVT transgenic mouse was provided by Dr. Shang Cai from Westlake University, male C57BL/6J mice were provided by Westlake University Animal facility. One female MMTV-PyVT transgenic mouse (3-month old) was used for this study as well as three male APP/PS1 (4-month old), three male APP/PS1 (18-month old), three male C57BL/6J (4-month old), and three male C57BL/6J (18-month old). No statistical methods were used to estimate sample size for animal studies throughout. All mice were housed at strict barrier facilities with macroenvironmental temperature and humidity ranges of 20–26 °C and 40–70%, respectively. Food and water were provided ad libitum. Mouse rooms had a 12 h light/12 h dark cycle. The housing conditions were closely monitored and controlled. No statistical methods were used to predetermine sample size.

The mice were deeply anesthetized with 1% sodium pentobarbital, transcardially perfused with 1x phosphate-buffered saline (PBS), followed by 4% paraformaldehyde (PFA; Electron Microscopy Sciences, the United States). Animal perfusion for all experiments was performed in random order. For all experiments, the experimenters were not blinded during allocation of animals to each group because we had to make sure that each group contained equal number of animals with matching ages according to the experimental design. Tissues were harvested and postfixed in 4% PFA for 12 h at 4 °C. The collected mouse brain, mouse liver, and breast cancer tissues were rinsed in 1x PBS and sectioned at 30 μm by VT1200S Vibratome (Leica VT1000S, Germany). Lung samples were prepared using cryostat (Leica CM3050S, Germany) following standard protocol. The proExM-MS and ProteomEx comparison was performed on the 30-μm thick brain slices. Coronal tissue sections from the midline of the mouse brain containing the hippocampus, cross-sections of breast cancer, and sagittal sections of mouse liver tissues were selected for further processing and imaging.

### Immunohistochemistry

Briefly, free-floating brain sections were treated with blocking buffer containing 5% bovine serum albumin (BSA, Beyotime, China) in PBS with 0.3% Triton X-100 (PBST) for one hour at 23 °C. Then brain sections were incubated with primary rabbit monoclonal antibodies for mouse anti-β amyloid (D54D2) XP® (1:1,000; 8243S; Cell Signaling Technology, US) diluted with 1% BSA in PBST overnight at 4 °C. After washing three times with PBS, sections were further incubated with secondary antibody (Alexa Fluor® 647 Goat Anti-Rat IgG, 1:1200; ab150167; Abcam, US) at 23 °C. The sections were then washed three times with PBS and counterstained with DAPI (5 μM in PBS; ab228549; Abcam, US), before being mounted with Prolong Gold antifade reagent (ThermoFisher; US).

### Tissue expansion and staining

The composition and storage conditions for all reagents and buffers used for tissue expansion and staining are described in Supplementary Table 5. Suppliers and lot numbers for chemicals and reagents used for ProteomEx are listed in Supplementary Table 6. PFA fixed tissue sections were briefly washed in PBS and treated with BT buffer at 23–25 °C for 2 h. Then the samples were briefly washed three times with MES buffer (pH = 6.0) and incubated with protein anchoring solution (0.1 mg/mL NSA in 100 mM MES at pH = 6.0) at 23–25 °C for 12 h. To remove the anchoring solution, the samples were washed with anchoring stopping buffer (100 mM MOPS at pH = 7.0) three times for 5 min each time. For gelation NSA-anchored tissue was incubated with Activated Monomer Solution ATMS (ATMS; Monomer Solution with APS and TEMED) in gelation chamber at 4 °C for 12 h and then transferred to vacuum oven (DZF-6000, Shanghai Sunrise Instrument) for polymerization reaction at 37 °C in $N_2$ atmosphere for 3.5 h. The formed tissue-hydrogel composite was removed from the gelation chamber, transferred to a dish, and submerged into homogenization buffer at 95 °C for 3 h. During this step, the gel can expand up to 3-fold in linear dimension. Homogenized samples were transferred to a bigger dish and washed 3 times with 10 mL of 50 mM Tris (pH = 8.8) for 30 min per wash at 23–25 °C. For washing steps, gel samples can be incubated on a shaker or rocker to facilitate solution exchange.

For Coomassie staining, first gels were washed 3 times with 50% methanol for 15 min per wash and 1 time with 100% methanol for 1 h at 23–25 °C. Methanol was replaced with fast Coomassie stain solution and incubated for 1.5 h at 95 °C then replaced with fresh fast Coomassie stain solution and incubated for another 2.5 h at 95 °C. During Coomassie staining, gel may shrink to the original size or smaller, however it did not influence the mechanical stability of the hydrogel in the expanded state. For gel destaining, samples were serially washed in 50 mM Tris, 25 mM Tris, 10 mM Tris and 2.5 mM Tris (all buffers pH = 8.8) for 1 h each at 60 °C. After the last washing step, gels were kept in 2.5 mM Tris pH = 8.8 at 23–25 °C until imaging and microdissection.

During washing steps, gels were gradually expanding reaching a linear expansion factor of 5.5- to 8-fold. Imaging of tissue before and after the expansion was performed using Zeiss Fluorescence Stereo Zoom Microscope (Axio Zoom.V16) in brightfield mode controlled by ZEN 3.1 software.

Tissue expansion for proExM-MS was performed according to the previously published protocol[9]. Briefly, PFA-fixed mouse brain tissue was treated with succinimidyl ester of 6-((acryloyl)amino) hexanoic acid (AcX), 0.1 mg/mL in PBS overnight in a humid chamber at 22 °C. Freshly prepared monomer solution (8.6% (w/v) sodium acrylate, 30%(v/v) acrylamide/bisacrylamide (30% solution; 37.5:1), 2 M NaCl, 0.01% (w/v) 4-hydroxy-2,2,6,6-tetramethylpiperidine-1-oxyl (4-hydroxy TEMPO) inhibitor dissolved in 1x PBS and supplemented with 0.2% (w/v) TEMED, and 0.2% (w/v) APS) was deposited on the tissue slice and evenly spread inside a gelation chamber then covered with glass slide and incubated in vacuum oven (DZF-6000, Shanghai Sunrise Instrument) at 37 °C for 2 h. Homogenization was performed with 5% SDS in water incubated at 58 °C in a humidity chamber overnight. For expansion, gels were rinsed in water and placed in excess volumes of doubly deionized water for ~1 h to expand while exchanging water every 15 min.

### Proteomic sample preparation from tissue-hydrogel composite

Manual microdissections were performed using commercially available biopsy punches (1 mm, 2 mm, 3 mm, 4 mm, 5 mm; Integra Miltex, USA) from target areas of the expanded Coomassie-stained samples. The excised tissue-hydrogel samples were washed and rehydrated in ddH$_2$O three times at 25 °C and incubated with 50%/50% (v/v) acetonitrile (ACN)/ddH$_2$O for 30 min at 30 °C on a shaker to remove residues of Coomassie staining. Samples were then washed in 50%/50% (v/v) ACN/100 mM ammonium bicarbonate (ABB) for 10 min and dried out in SpeedVac under 45 °C for 3 min. The dried samples were treated with 20 mM TCEP for 30 minutes in darkness at 32 °C followed by the alkylation step by adding 55 mM iodoacetamide (IAA) and incubating for 30 minutes in the dark at 25 °C. Samples pieces were further washed with 100 mM ABB. Samples were dehydrated by washing with 50%/50% (v/v) ACN/100 mM ABB 2 times for 5 minutes each at 25 °C on a shaker and then dried out in SpeedVac. Protein digestion was performed with 12.5 ng/μL trypsin (Hualishi Tech. Ltd, Beijing, China) in 25 mM ABB (pH = 8.0) incubating twice at 37 °C for 4 h and 8 h. For the experiments in Fig. 4, experimenters were blinded to tissue samples collection, preparation, and MS data acquisition.

Digested peptide solutions were collected in the following steps and combined: 1) collect 30–40 μL of the supernatant; 2) add 100 μL 25 mM ABB, vortex for 10 min at 25 °C and collect supernatant; 3) add 100 μL 50% ACN/2.5% formic acid and vortex for 10 min, collect supernatant and repeat three times; 4) add 100 μL 100% ACN and vortex until the gel pieces turning white and sticky. Peptide samples were placed under vacuum to reduce the volume to 20–30 μL. The peptides were then desalted using C18 spin columns (Pierce™ C18 Spin Tips, Thermo Fisher Scientific, US) and dried in a SpeedVac. The cleaned peptides were stored at –20 °C until further analysis.

### Proteomic sample preparation by PCT

Mice brain slices were weighed and processed via the accelerated PCT workflow[42]. Briefly, each slice was transferred into a PCT-MicroTube (Pressure Biosciences Inc., MA, US) and hydrolyzed with 100 mM Tris-HCl (pH = 10) for 30 min at 95 °C. The PCT-MicroTube was immediately cooled on ice after basic hydrolysis. Lysis buffer (6 M urea, 2 M thiourea), 20 mM TCEP, and 40 mM IAA were added to the PCT-MicroTube. The PCT scheme for tissue lysis was 90 oscillating cycles, each with 45,000 psi for 30 s and ambient pressure for

10 s at 30 °C using the Barocycler NEP2320-45K (Pressure Biosciences Inc., MA, US). Then, the denatured proteins were digested using a mixture of trypsin and Lys-C (Hualishi Tech. Ltd, Beijing, China) with an enzyme-to-substrate (w/w) ratio of 1:20 and 1:80, respectively. The PCT scheme for protolysis consisted of 120 cycles at 30 °C, with 50 s of high pressure at 20,000 psi and 10 s of ambient pressure for each cycle. Trifluoroacetic acid was added to the solution at a final concentration of 1% to stop enzymatic digestion. Peptides were desalted by the C18 96-well plate (Thermo Fisher Scientific, US) and dried by a SpeedVac (Thermo Fisher Scientific, US). The dried peptide fractions were reconstituted in 0.1% formic acid.

### High-pH reversed-phase chromatography fractionation

Approximately 100 μg mouse brain pooled peptides from AD and WT mouse were fractioned on a chromatographic column (BEH C18, 300 Å, 5 μm, 4.6 mm ×250 mm) coupled to a Thermo Dinex Ultimate 3000 (Thermo Fisher Scientific, US) within a 120 min effective gradient of 5–35% buffer B (98% ACN, 0.6% ammonia, pH = 10) and separated into 120 fractions separated by 1 min interval. The fractions were further combined into 30 samples. All samples were dried by a SpeedVac (Thermo Fisher Scientific, the United States), reconstituted in 0.1% formic acid, and spiked with standard peptides (iRT; Biognosys, Switzerland).

### Liquid chromatography

The peptides and fractioned peptide samples were separated on a 15 cm × 75 μm silica column custom packed with 1.9 μm 100 Å C18 aqua installed into nanoElute® system (Bruker Daltonics, Bremen, Germany). The mobile phase was mixed with buffer A (2% ACN, 0.1% formic acid) and buffer B (98% ACN, 0.1% formic acid). The buffer B (%) was linearly increased from 5 to 27%, followed by an increase to 40% within 10 min and a further boost to 80%. Effective linear gradients of different duration were performed. The durations of DDA for spectral library construction, DDA for benchmark study, and PulseDIA for application study were 50 min, 80 min, and 50 min * 2 Pulse runs, respectively.

### Proteomic data acquisition by DDA mode

Eluted peptides were analyzed in a hybrid trapped ion mobility spectrometry (TIMS) quadrupole time-of-flight mass spectrometer (timsTOF Pro, Bruker Daltonics, Bremen, Germany) via a CaptiveSpray nano-electrospray ion source. MS data was acquired using the mass spectrometers control software Bruker otofControl v6.2 and HyStar v5.1. DDA was performed in PASEF mode with 10 PASEF scans per topN acquisition cycle for the ion mobility-enhanced spectral library generation[43]. The accumulation and ramp time was 100 ms each for a dual TIMS analyzer, achieving a total cycle time of 1.17 s. The ion mobility was scanned from 0.6 to 1.6 Vs/cm$^2$. MS1 and MS2 acquisition was performed in the range of m/z from 100 to 1700 Th. Precursors that reached a target value of 20,000 arbitrary units were dynamically excluded for 0.4 min. Singly charged precursors were excluded by their position in the m/z–ion mobility plane. Peptide extraction and MS data acquisition for Fig. 4 were conducted in random order to minimize batch effect.

### Proteomic data acquisition in a PulseDIA mode

The data-independent acquisition Parallel Accumulation Serial Fragmentation (diaPASEF) was performed in a PulseDIA mode[43]. The other setting was the same as DDA mode. The ion mobility was scanned from 0.7 to 1.3 Vs/cm$^2$. MS1 and MS2 acquisition was performed in the range of m/z from 100 to 1700 Th. We defined two sets of complementary isolation windows (Supplementary Table 7) and applied them to two MS methods for two injections[43]. Peptide extraction and MS data acquisition for Fig. 4 were conducted in random order to minimize batch effect.

## Proteomic data analysis and mouse brain-specific spectral library generation

The DDA data were analyzed using FragPipe (version 15.0) platform with the MSFragger (version 3.1.1)[44,45] search engine against a FASTA file from SwissProt containing 17,282 mouse protein entries and 17,282 decoys sequences. The workflow "SpecLib" was used. The "IN-MS" was selected as input LC/MS file. Fragment mass tolerance was set by 0.05 Da. Digestion enzyme was trypsin with cutting after "KR" but not before "P". The self-constructed library was further used in PulseDIA data analysis by DIA-NN (version 1.7.15)[46]. False discovery rates of precursors and proteins were set at 1%. Other settings were used as default parameters.

## Proteomic data quality control analysis

Experimenters were blinded during peptide and protein identification step. Experimenters were not blinded during bioinformatic analysis because animal groups had to be compared together. In the application section, we processed 144 hydrogel samples from the 12 mouse brain slices in parallel by ProteomEx protocol. Eight samples were lost during the peptide extraction step and thus were not subjected to MS analysis. The 136 samples were allocated into 12 batches for proteomic data acquisition and each batch contained one mouse brain pooled sample as quality control. The stability of the mass spectrometer was confirmed by coefficient variation of protein abundance on pooled samples, which was less than 0.15 (Supplementary Fig. 11). 14 samples with fewer than 1464 protein identifications were excluded from downstream data analysis. As expected based on the genotype, protein expression abundance of APP and PSEN1 were significantly dysregulated in Young WT and Young AD. In old groups, the expressions of APP and PSEN1 were upregulated in the old AD group with $P$-values of 0.056 and 0.026, respectively (Supplementary Fig. 11).

## Isotropic expansion analysis

To perform isotropic analysis of tissue expansion we used a B-spline-based image registration MATLAB package to register post-ExM to pre-ExM image by landmark selection. The input images for the script were converted to grayscale and downsampled to be smaller than 2000 by 2000 pixels to save computational time. The pixel resolution was manually input into the script for error measurement. The pre-ExM image was set as a static image, and the post-ExM image was set as a moving image to be registered. First, for rigid registration, five landmarks were manually selected and annotated pair wisely in pre- and post-ExM images. After the rigid registration, twenty landmarks were manually selected over all parts of the tissue sample. The script computed nonrigid registration, refinement, deformation vector field, root mean square error plot, and the overlay of pre- and registered post-ExM image. The compiled RMS plot with standard deviation for each tissue type was computed from the error measurement data in saved workspaces of all analyzed samples.

## Data analysis and statistics

The images were analyzed with ImageJ (version 1.53f51) and Zeiss ZEN 3.1 software. Brain tissue annotations were performed using the DeepSlice online app (https://www.deepslice.com.au/) and the QuickNII VisuAlign software. MS data were analyzed using FragPipe platform (version 15.0) with the MSFragger (version 3.1.1) and the DIA-NN software (version 1.7.15). Statistical charts are drawn with PRISM (version 9.2.0), R (version 4.0.3), Python (version 3.7) and MATLAB (version R2021a). Isotropic analysis calculation performed with MATLAB (version R2021a). Missing values were omitted when we calculated the fold change. In the t-SNE analysis, missing values were imputed by 0.8*min intensity in the matrix of 6215 proteins. The differentially expressed protein (DEP) comparison of AD vs. WT was performed at a threshold of adjusted $P$-value <0.05 and fold change >2 ($P$-values were calculated by two-tailed Student's $t$ test and adjusted by the method of Benjamini–Hochberg). GO term enrichment was conducted by Metascape web server (http://metascape.org/) using default parameters. Pathway enrichment and biomarkers identification were performed by IPA, in which the $P$-value was estimated by right-tailed Fisher's exact test. We did not perform a power analysis for sample number, since our goal was to develop a new technology; as noted in ref. 47, and recommended by the NIH, "In experiments based on the success or failure of a desired goal, the number of animals required is difficult to estimate…" As noted in the aforementioned paper, "The number of animals required is usually estimated by experience instead of by any formal statistical calculation, although the procedures will be terminated [when the goal is achieved]." The number of samples used reflect our past experience in developing biotechnologies.

## Reporting summary

Further information on research design is available in the Nature Portfolio Reporting Summary linked to this article.

## Data availability

The mass spectrometry proteomics data generated in this study have been deposited to the iProX database with the dataset identifier IPX0003949000. Raw data including raw images essential to the work are available online as Source Data file and provided in the Supplementary Information. The complete datasets for Figs. 3A, B, 4E, Supplementary Fig. 11D including raw images are available at FigShare [https://doi.org/10.6084/m9.figshare.21431157] and Zenodo [https://doi.org/10.5281/zenodo.7266442]. Source data are provided with this paper.

## Code availability

Code is available under the BSD-2-Clause license on GitHub [https://github.com/lilulu777/ProteomEx] and Zenodo [https://doi.org/10.5281/zenodo.7266442].

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

## Acknowledgements

We thank Rujie Qi and Yuan Yao from Westlake University for help with characterization of hydrogel mechanical stability. We also thank Xun Guo from Westlake University for help with tissue sample preparation. Figures 1A and 4A were created with Biorender.com. This work was supported by National Key R&D Program of China (No. 2021YFA1301602, 2021YFA1301601, 2020YFE0202200), the Zhejiang Provincial Natural Science Foundation for Distinguished Young Scholars (LR19C050001), Hangzhou Agriculture and Society Advancement Program (20190101A04), National Natural Science Foundation of China (81972492) all to T.G., and start-up funding from the Foundation of Westlake University, projects 32050410298 and 32171093 supported by the National Natural Science Foundation of China, MRIC Funding 103536022023 all to K.D.P. This work was further supported by the grant from National Key R&D Program of China (No. 2022YFF0608400, 2022YFF0608403) to Y.Zhu. This work was also supported by grants from China Postdoctoral Science Foundation (2021TQ0283) and International Postdoctoral Exchange Fellowship Program (Talent-Introduction Program, YJ20210170) to Z.D. We thank the Research Center for Industries of the Future (RCIF) at Westlake University for partially supporting this work.

## Author contributions

L.L., C.S., Y.S., T.G., and K.D.P designed the experiments. X.C., S.C., and H.X. provided the animals. H.Z. and X.S. prepared fixed tissue samples. C.S. and K.D.P. developed the ProteomEx hydrogel. C.S. developed the

Coomasie staining procedure. L.L., Y.S., Y.Zhu developed optimized in-gel digestion protocol. L.L., Y.S., Z.D., and R.W. performed proteomic experiments. L.L. and W.J. participated in MS maintenance and proteomic data acquisition. L.L., Y.S., R.W., Y.Zhou, and Z.D. conducted proteomic data analysis and R.W. performed the isotropic analysis. L.L., C.S., Y.S., and K.D.P wrote the manuscript, and the others revised the manuscript. K.D.P and T.G. conceived the project and oversaw all its aspects.

## Competing interests

T.G. and Y.Zhu are shareholders of Westlake Omics Inc. L.L., C.S., Y.S., Y.Zhu, T.G., and K.D.P are listed as co-inventors on a patent covering the ProteomEx technology filed by Westlake University (patent application PCT/CN2022/115456 and CN patent application 202211042411.8). The remaining authors declare no competing interests.
