## [Peer Review File · Nature Communications]

Spatially resolved proteomics via tissue expansionREVIEWER COMMENTS

Reviewer #1 (Remarks to the Author):

Reviewer expertise: LC-MS based proteomics, cancer tissue proteomics

In the current study, Li and colleagues describe a method for spatially resolved tissue proteomics by a combination of tissue expansion methodology with unbiased, LC-MS based proteomics, called ProteomEx. This concept is benchmarked and applied to different tissue types from brain, liver and breast cancer as well as to a mouse model of Alzheimer's disease. While a conceptually similar publication was published last year (Drelich et al, 2021), the authors provide explanations and side-by-side comparisons how their method differs and represents an improved pipeline. For instance, the authors report higher peptide yields and spatial resolution, increased peptide and protein identifications, as well as compatibility with different staining procedures. Moreover, compared to Drelich et al, the authors include more biological use cases to demonstrate the biomedical utility of their approach.

Overall, the manuscript is well written and the story easy to follow. Figures are clearly presented and the developed pipeline represents an innovative approach for spatial tissue proteomics, which should be of interest for the readership of Nature Communications. However, there are several key questions raised by the results that should be addressed prior publication.

The authors claim that their approach was validated and applied to high-throughput large-scale proteomics studies. The workflow introduced in Fig. 1 suggests a very cumbersome procedure involving many manual processing steps from sample preparation until mass spectrometric acquisition and data analysis. Here, the authors could provide time scales to assess how streamlined this workflow in-fact is. Regarding throughput, how many tissue slices can be processed in parallel? If high-throughput rather refers to the number of regions collected and processed per single slice, the analysis of in total 144 gel slices manually punched out from regions of interest combined with manual pipetting should not be considered a high-throughput methodology. Similarly, based on the data provided, I do not see how the ProteomEx approach would have any higher throughput than classic LCM approaches, which the authors introduce as low throughput method. In LCM, ROIs are isolated by an UV laser instead of punches and directly transferred into 96-well collection plates without the need to do any tissue expansion. I recommend to tone down this high-throughput statement in the abstract and discussion.

More specific points:

1. In Fig 2, the authors benchmark their ProteomEx approach. The data provided is mainly of qualitative nature and an important analysis that is missing here is about the quantitative reproducibility of their approach. At a minimum, scatter plots of replicates from adjacent microregions should be shown including proteome correlation values. How does this compare to the other sample processing protocols from Fig 2?
2. How does the anchoring and embedding procedure affect peptide modifications? MSFragger could be used in open-search mode to compare differences in peptide modifications in comparison to in-solution and PCT based bulk preparations where no anchoring and embedding is applied.
3. For the immunolabeling data (Fig. S4), control stains are missing to assess the labeling specificity. The authors should provide positive and negative tissue staining controls for the chosen antibody.
4. In Fig 4, information on data completeness and the applied data filtering strategy is missing. Was data imputation performed? A higher number of missing data in one or more sample groups could affect the t-SNE results. In table 4C, a higher number of proteins quantifications was observed for old animals. Can the authors comment on this and provide more details on how this was taken into account when analyzing the data?
5. In supplementary figure 7, the comparison to other spatial proteomics approaches lacks important information to interpret this figure. Please also provide information on the different acquisition strategies (dda and dia), instrument types, software used and LC gradient lengths. A table would be helpful for this comparison.
6. The authors mention the limited application of ProteomEX to tissue regions smaller than 125µm

(page 15) but then highlight the integration with super-resolution microscopy in the discussion. How would this fit together?

Reviewer #2 (Remarks to the Author):

The manuscript titled "Spatially Resolved Proteomics via Tissue Expansion" describes a method coupling tissue expansion with downstream spatial proteomics analysis. The dissection of spatially resolved tissue is performed using biopsy punches. The methods described here can certainly allow many laboratories with limited resources to explore spatially resolved proteomics as an option. The manuscript is well written and should be well received by the proteomics community.

However, a few important technical details are missing. Additionally, the authors tend to misleadingly over-represent their work as superior to all other methods. Their method sounds great and useful, and I anticipate a good adoption, nevertheless, in many aspects, it is not superior to some of the methodologies discussed in the paper. The authors should not only present the advantages of their method but also discuss its limitations.

I hope that the following comment will help the authors to improve the quality and the impact of their manuscript.

Minor comments:

Line 63 the authors indicate that "data-independent acquisition (DIA) MS offers superior depth and throughput of proteomic analysis." without precisizing over what. Looking at the methods presented, the sample preparation here will longer than the 12 hours of digestion and the runs are of an average duration of their runs is 2 pulseDIA of 50 min/sample which is longer than their DDA benchmark study. Every claim made in the paper stating that the method is faster than other methods is misleading.

An example of misleading statement (L68-71) is the following: "Several techniques based on these approaches can achieve up to $\sim 50 \mu\text{m}$ of lateral resolution on thin (10's of μm) tissue sections 20, however, they require special equipment not accessible to most labs. In addition, they are technically fragile and challenging in implementation, and have limited throughput and protein profiling depth."

Only the NanoPOTS imaging method is cited here when the authors indicate that "several techniques" are available for such resolutions. They describe alternate methods as technically fragile without explaining why (this sounds more like an opinion than a reality). Similarly, the authors are indicating that LCM methods require special equipment not accessible to most labs, Mass spectrometry is another example of equipment that is not accessible to most lab, is expensive and require extensive expertise, yet their method utilize mass spectrometry. They state that alternate methods "have limited throughput" while the throughput of the example chosen (NanoPOTS) seems very similar to the ProteomEx one. As pointed by the authors in the evaluation of ProteomEx, the protein depth depends on the volume of tissue analyzed. By comparing the depth obtained from the smaller voxels $50\mu\text{m} \times 2 \times 10\mu\text{m}$ to the much larger voxels of ProteomEx the authors mislead the reader to believe that ProteomEx is superior to other spatially-resolved methods altogether.

Many other statements of this type across the paper would need to be tempered (examples : L332, 374, L229-334)

Major comments:

The authors have not indicated the thickness of the sections they employed with their method. This is critical as they compare their method to the published one by Drelich et al. by simply comparing the peptide and protein counts to those obtained by this group. If the authors have

used thicker brain sections (e.g., 16 μm) this could explain why their "peptide yield and protein identification" are better and may have nothing to do with the slight difference in employed technologies.

L204 the other list the organs "brain, liver, and breast cancer" those are all "full" tissues. It would be great to evaluate potential delocalization for hollow organs such as Lungs for example as we can expect those to be more prone to deformation due to the empty space they contain.

Figure 2G-H and L192-195 the authors state "Albeit it was more peptides and proteins identified by PCT due to a larger sample injection amount, ProteomEx showed a higher degree of reproducibility in the processing of small sample volumes compared to PCT (Figure 2G, H). ProteomEx provides a new strategy for sub-nanoliter volume sample preparation for proteomic analysis." When looking at the figure it seems clear that there is a single outlier for PCT that resulted in what the authors call "higher variability". The punch cuts were realized on non-homogenous tissues and it is impossible to rule out that this specific cut was not done in a less dense region (e.g. overlapping with the exterior of the tissue) than all the other cuts. From the data presented without additional metrics, it is impossible to evaluate if PCT or ProteomeEx reproducibility is any different. The evaluation of the technical reproducibility has to be done on a homogenous tissue deposition rather than on a heterogenous tissue. Real metrics could then be discussed regarding the reproducibility of the method (CVs, sample-to-sample correlations, percentage of peptides/proteins identified in multiple runs, etc.).

The authors admit (L320) that "manual handling of small transparent gel samples, which were hard to see by the naked eye" limit the resolution to which samples can be obtained. The size of the tissue disks obtained is constrained by the availability of punch cutters. I would be very surprised to see how the method can be implemented to cut precisely with the naked eye 50 single glomerulus of 150 to 300 μm of diameter to perform single glomerulus proteomics. LCM technologies are compatible with immunofluorescence for the identification of specific cells/tissues organization and enable a much more precise dissection of tissues. The ProteomeEx method can only be used to perform spatially resolved sections on relatively homogenous tissue regions. This major limitation among other that can be noted should be discussed in the manuscript.

Reviewer #3 (Remarks to the Author):

Summary:

This work combined the ability of hydrogel expansion with mass spectrometry based proteomics. The authors showed a modified gel recipe with improved expansion factors, Coomassie staining of the expanded hydrogels and microdissection/ microdigestion protocols. As mentioned and discussed by the authors, this paper conceptually overlaps with Drelich's work [1] from last year, in which the expansion combined with mass spectrometry based proteomics was also shown. The authors mentioned in the manuscript that the two researches differ mainly in 1) Expansion protocol, 2) higher peptide yield and 3) accurate microdissection with the colorimetric staining, and performed side-by-side comparison.

There are still some general questions that might need to be clarified:

1. The expansion factor (EF) mentioned in the paper is from 5.5 to 8 with tissue embedding and the test of new gel composition is with final expansion factor of an empty gel from 5.3 to 8.4. And in the text, there was one EF = 8 image shown in S4, the rest analyzed data or images was however with only lower magnification. This is relatively big variance compared to the other ExM techniques. Is that due to monomer quality difference between batches or from some other issue, such as incomplete homogenization? And if most of the experiments are not characterized with the 8x protocol (e.g. peptide retention, distortion analysis), it would be appropriate to change the discussion/calculation in the maintext with the more reproducible expansion factor.

2. The term "reversible protein anchoring to polymer" is used to describe N-Succinimidyl acrylate, but I didn't find why it is "reversible". Although NSA was used in ExM for the first time, a very similar chemical Methacrylic acid NHS ester was published since 2016 and used in tissues ExM in multiply labs ([2],[3]). As my understanding this chemical is even cheaper, also relatively stable. Did the authors already compared this anchoring molecule MA-NHS with the NSA?
3. The distortion is not as important in this study compared to optical based ExM. But the distortion is still relatively larger (10%) compared to many of the published expansion protocol for tissues (5% or less). This will still significantly limit the new recipe to be broadly used in microscopy related studies. Did the authors maybe also perform some distortion analysis during the gelation composition selection? Or can this gel composition be further improved for a lower distortion over long distance?
4. It was mentioned in the SI that reducing the concentration of crosslinker will result in more fragile gels that may not show uniform expansion. There was a new study [4] that proposed the modification with low crosslinker concentration, better handling and near 10x expansion with the conventional chemicals. The author might need to add the paper as a reference.
5. The authors claimed the screening of 400 hydrogels with the criteria expansion factor and mechanical stability. Current ExM are mostly based on sodium acrylate, but it was completely filtered out. If sodium methacrylate is significantly better, it would be better if the example of being mechanically stable can be shown as video or figures.

Some small points:

1. It was mentioned in the SI note 1, "We ultimately selected the two hydrogel recipes consisting of SMA:DMAA:PAE in molar ratio 1:4:0.0008 and SMA:DMAA:TPT in molar ratio 1:4:0.0005, which were characterized by linear expansion factors of 8.2 and 8.4, respectively." But in the SI table 1, only 1:4:0.0008 was labelled as a ProteomEx.
2. The coomassie staining part was quite interesting, which facilitates the precise cutting, but also requires more handling on the gel, staining and destaining. I wonder if the destaining step or treatment with methanol etc would influence the stiffness of the hydrogel.
3. The part for in-gel digestion is also very nice. LysC and trypsin was previously used for homogenization, here the authors cleverly used them for peptide extraction.
4. The proteome analysis seems to be based on the structures identified by naked eye with coomassie staining. By how much would the local distortion or heterogeneous expansion in hydrogel influence the proteome analysis result?

[1] Drelich et al. *Acs anal. chem*, 2021

[2] Fan et al., chapter 7 in *Expansion Microscopy for Cell Biology* (2021)

[3] Chozinski et al. *Nat. Methods*, 2016

[4] Damstra et al. *Elife*, 2022

We thank Reviewers and Editors for the valuable feedback and comments that helped us improve the manuscript. According to the suggestions, we performed additional experiments and data analysis to improve the thoroughness and fairness of the method comparison. In particular, we used liver tissue to validate technical reproducibility and stability of the ProteomEx method for microsample processing compared to PCT. We carried out isotropic analysis for mouse lung tissue and measured mechanical stability of hydrogels. The newly obtained datasets as well as raw datasets acquired before were used to perform additional analysis to evaluate quantitative and technical reproducibility, labeling specificity, PTM recovery. The manuscript and Figure 1 were edited to provide detailed description of the ProteomEx workflow indicating duration of each step. According to Reviewers suggestions we revised claims and conclusions to be more precise and avoid misleading over-representation of the ProteomEx technique, we also cited the papers suggested by Reviewers.

Altogether, we have added five new Supplementary Figures, one Supplementary Note, one Supplementary Table, we modified Figure 1 by adding workflow chart for the ProteomEx procedure. We also revised the main text to describe all newly added results and tempered claims that Reviews found misleading or imprecise. The newly added and edited text is highlighted with blue. We believe that we addressed all comments and concerns in full. Please find below detailed point-by-point responses to the comments from Reviewers.

Reviewer #1 (Remarks to the Author):

Reviewer expertise: LC-MS based proteomics, cancer tissue proteomics

In the current study, Li and colleagues describe a method for spatially resolved tissue proteomics by a combination of tissue expansion methodology with unbiased, LC-MS based proteomics, called ProteomEx. This concept is benchmarked and applied to different tissue types from brain, liver and breast cancer as well as to a mouse model of Alzheimer's disease. While a conceptually similar publication was published last year (Drelich et al, 2021), the authors provide explanations and side-by-side comparisons how their method differs and represents an improved pipeline. For instance, the authors report higher peptide yields and spatial resolution, increased peptide and protein identifications, as well as compatibility with different staining procedures. Moreover, compared to Drelich et al, the authors include more biological use cases to demonstrate the biomedical utility of their approach.

Overall, the manuscript is well written and the story easy to follow. Figures are clearly presented and the developed pipeline represents an innovative approach for spatial tissue proteomics, which

should be of interest for the readership of Nature Communications. However, there are several key questions raised by the results that should be addressed prior publication.

We thank Reviewer for appreciating our work and for providing the positive and helpful comments, which we have addressed in full below.

The authors claim that their approach was validated and applied to high-throughput large-scale proteomics studies. The workflow introduced in Fig. 1 suggests a very cumbersome procedure involving many manual processing steps from sample preparation until mass spectrometric acquisition and data analysis. Here, the authors could provide time scales to assess how streamlined this workflow in-fact is. Regarding throughput, how many tissue slices can be processed in parallel? If high-throughput rather refers to the number of regions collected and processed per single slice, the analysis of in total 144 gel slices manually punched out from regions of interest combined with manual pipetting should not be considered a high-throughput methodology. Similarly, based on the data provided, I do not see how the ProteomEx approach would have any higher throughput than classic LCM approaches, which the authors introduce as low throughput method. In LCM, ROIs are isolated by an UV laser instead of punches and directly transferred into 96-well collection plates without the need to do any tissue expansion. I recommend to tone down this high-throughput statement in the abstract and discussion.

We thank the Reviewer for the great suggestion. We agree that visualization of the timeline can help assess the ProteomEx workflow. Correspondingly, we modified **Figure 1** to add the ProteomEx timeline indicating the total duration of each step as well as the hands-on time required for the procedures. While it takes about 58 hours to process fixed samples, the required hands-on time is only 5 h, i.e., less than 10% of total duration. Since major manipulations in the ProteomEx workflow are adding and removing buffers and reagents, multiple samples can be processed in parallel including in-gel digestion. For example, all samples for **Figure 4** (12 brain slices and 144 gel samples) were processed in parallel within 58 hours starting from fixed brain slices. However, the manual microdissection is a bottleneck in this protocol as expanded samples have to be processed one by one manually. Therefore, we agree with the Reviewer that “high-throughput” term might be misleading in the context we used in the original manuscript. We have revised the manuscript to refrain from using the term “high-throughput” for ProteomEx as well as “low-throughput” term in regard to LCM-based methods. We also revised the Discussion section to discuss the ProteomEx timeline. Please see revised **Figure 1** and Discussion section P. 15 L. 320-322, P. 15-16 L. 338-342.

More specific points:

1. In Fig 2, the authors benchmark their ProteomEx approach. The data provided is mainly of qualitative nature and an important analysis that is missing here is about the quantitative reproducibility of their approach. At a minimum, scatter plots of replicates from adjacent microregions should be shown including proteome correlation values. How does this compare to the other sample processing protocols from Fig 2?

We thank Reviewer for this great comment. To address this comment, we reanalyzed our DDA files from the four methods (In-solution, PCT, proExM-MS, ProteomEx) using the MSFragger

software in the label-free quantification (LFQ) mode. We presented the Pearson correlation between each pair of samples as a heatmap and coefficient of variation (CV) values were visualized using violin plots (see the Figure below). The gel punches from the adjacent microregions sampled by ProteomEx (5.9 nL) were also added for comparison.

The Pearson correlations of the replicates from the same methods were mostly greater than 0.90, except for one sample prepared by the in-solution strategy, which showed a higher deviation than the other methods (**Supplementary Figure 6**). The samples processed with ProteomEx achieved relatively high Pearson correlation values of about 0.94-0.96 in the quarter of the slice (250 nL) group. The correlation values of the microregion samples (5.9 nL) were in the range of 0.89-0.92, which was lower than the values of the quarter of the slice (250 nL) most likely due to the heterogeneity of the mouse brain tissue. Overall, these values were higher or comparable to the other three methods used for benchmarking, *i.e.*, 0.89-0.95 for proExM-MS, 0.90-0.95 for PCT, and 0.81-0.96 for in-solution. The median CV of protein intensity (red point in the violin plots) is the lowest for the quarter of slice samples prepared with ProteomEx, which indicated the highest reproducibility of this method (see below). New results were added to the manuscript, please see the revised main text P. 9 L. 184-192 and the newly added **Supplementary Figure 6**.

Supplementary Figure 6. Reproducibility and stability comparison for the selected sample preparation methods. (A) Heatmap of Pearson correlations for protein quantification for each paired samples from the four sample preparation methods analyzed using the MSFragger software (n=4, 4, 7, 4, 3 technical replicates from one, one, two, one, and one brain slices for in-solution digestion, PCT, proExM-MS, ProteomEx, and ProteomEx (5.9 nL sample), respectively; the raw datasets corresponding to Figure 2C were used for analysis). The color bar indicates the values of Pearson correlations. (B) Coefficient of variation of quantified protein abundance from the four methods.

2. How does the anchoring and embedding procedure affect peptide modifications? MSFragger could be used in open-search mode to compare differences in peptide modifications in comparison to in-solution and PCT based bulk preparations where no anchoring and embedding is applied.

Thank you for your suggestions on further ProteomEx validation. According to the suggestions, we reanalyzed the DDA files used for data presentation in **Figure 2C, D** using MSFragger. To verify the chemical modifications that can be potentially introduced during ProteomEx procedure as well as post-translational modifications (PTMs) of peptides, we set the variable modifications with anchor mass shift (Mass delta 54.0474 for ProteomEx corresponding to the modification with NSA anchor, 114.1656 and 168.2130 for proExM-MS corresponding to the modification AcX anchor) on four amino acids (namely K/Q/R/N), which are primary targets of chemical anchor modification. It appeared that all four methods had more than 0.4% of peptides fraction with chemical modification corresponding to the NSA anchor (mass delta 54.0474). The macrosamples processed with ProteomEx showed ~1% of peptides with mass delta of 54.0474, which was about twice higher than the fractions of chemically modified peptides obtained with the in-solution digestions and PCT methods (see newly added **Supplementary Figure 5B**). However, it should be noted that for the mass delta modification of 54.0474, we found three other different chemical modifications with the matching mass delta that are naturally occurred (see the following link for details

http://www.unimod.org/modifications_list.php?a=search&value=1&SearchFor=54.0474&SearchOption=Contains&SearchField=). These naturally occurring chemical modifications may interfere with the real ratio of the ProteomEx anchor modification quantification and cannot be sorted out by analysis. The other analyzed mass shifts did not exceed 0.124% at the peptide level for all analyzed samples (**Supplementary Figure 5B**).

Next, we analyzed the post-translationally and chemically modified peptides for the four methods by MSFragger using open-search mode. We discovered 158 (88.76%) overlapped types of peptide modifications from a total of 178 modifications for all four methods while there were no unique modifications for ProteomEx and five common modifications between in-solution and PCT (**Supplementary Figure 5C**). Furthermore, we conducted a quantitative analysis of peptide modifications to identify their hierarchical clustering. According to the clustering analysis, PCT was similar to the in-solution digestion, followed by ProteomEx. However, results for ProExM-MS were quite different from the other three methods in the quantification of modified peptides (**Supplementary Figure 5D**). These results demonstrated that ProteomEx does not introduce any

unique modifications to the peptides compared to in-solution digestion and PCT that can interfere with protein identification.

Supplementary Figure 5. Post-translationally and chemically modified peptide analysis. (B) Anchor-modified peptide analysis. Y-axis indicates the anchor-modified peptide fraction (%). The values on the top of the sections are mass shift delta, 54.0474 for ProteomEx corresponding to the modification with NSA anchor, 114.1656, and 168.2130 for proExM-MS corresponding to the modification AcX anchor. The values for each column represent the mean, error bars represent standard deviation. *P*-values are estimated by t-test (**P*<0.05, ***P*<0.01, ****P*<0.001, pairs without marked *P*-value are statistically non-significant, *i.e.*, *P*>0.05). (C) The Upset plot with the numbers of overlapped modification of peptides for the four methods. (D) The heatmap illustrating

the percentage of different modifications of peptides. Each row represents a type of peptide modification. The rows and columns are clustered by the hierarchical method.

We revised the main text correspondingly (P. 8 L. 162-168) and added new **Supplementary Figure 5** and corresponding **Supplementary Note 3**.

For the immunolabeling data (Fig. S4), control stains are missing to assess the labeling specificity. The authors should provide positive and negative tissue staining controls for the chosen antibody.

We added the control stain images of AD and age-matching wild-type mice with antibody staining in revised **Supplementary Figure 9** as new panels A, B, and C.

Supplementary Figure 9. Compatibility of ProteomEx with immunostained and DAPI stained mouse brain tissue. (A,B,C) Fluorescence images of mouse brain slice stained with (A) DAPI and (B) anti- β amyloid antibodies and (C) merged image (brain slice collected from wildtype mouse). (D,E,F) Fluorescence images shows the brain slice collected from 18-month old APP/PS1 mouse. White circles represented the punched locations used for MS analysis. (G) Brightfield images of the pre-expansion and (H) Coomassie-stained expanded mouse brain tissue section (LEF = 6.11-fold). (I) Number of peptide and protein identifications from 2.52 nL brain tissue acquired in by PulseDIA mode.

4. In Fig 4, information on data completeness and the applied data filtering strategy is missing. Was data imputation performed? A higher number of missing data in one or more sample groups could affect the *t*-SNE results. In table 4C, a higher number of proteins quantifications was

observed for old animals. Can the authors comment on this and provide more details on how this was taken into account when analyzing the data?

First, we performed data filtering according to the following strategy. We deleted the lower 10% of the samples with protein identifications at a threshold of 1464, which was significantly lower than identification in 1 mm-diameter punch brain samples (~1500) collected for all four groups of animals. As a result, 14 samples were removed from the protein matrix.

Next, the data imputation was performed during t-SNE analysis. For the t-SNE plot in **Figure 4B**, we removed 8 proteins with 100% missing values with the remaining 6215 proteins. The missing values were imputed with $0.8 \times \text{min intensity}$ in the matrix of 6215 proteins. We did not impute the protein matrix when performing the limma analysis for differentially expressed protein (DEP) identification. We revised the Methods section of the manuscript to describe the filtering strategy and data imputation. Please see P. 29 L.623-624.

In supplementary figure 7, the comparison to other spatial proteomics approaches lacks important information to interpret this figure. Please also provide information on the different acquisition strategies (dda and dia), instrument types, software used and LC gradient lengths. A table would be helpful for this comparison.

We have added a table that provides important information to interpret **Supplementary Figure 7** in the original manuscript. In the table, we included information on acquisition strategies (DDA and DIA), instrument types, software used, LC gradient lengths, as well as the dimension of the tissue samples, and the number of identified peptides and proteins. Please see the newly added **Supplementary Table 4** and revised **Supplementary Figure 12**.

6. The authors mention the limited application of ProteomEX to tissue regions smaller than 125 μm (page 15) but then highlight the integration with super-resolution microscopy in the discussion. How would this fit together?

Samples expanded using ProteomEx protocol can be stained with fluorescence dyes, such as, for example, DAPI, and/or antibodies, as homogenization with SDS-based buffer should preserve epitopes. This will enable super-resolution imaging of samples under conventional diffraction-limited microscopy (Tillberg et al., Nat. Biotech. 2016). Previously, it was shown that by using ExM it was possible to improve pathological diagnostics of kidney minimal change disease and classification of early breast cancer lesions (Zhao et al., Nat. Biotech. 2017). In this case, super-resolution imaging of tissue can pinpoint regions of interest for proteomic analysis. For example, visualization of podocyte structures, which requires super-resolution imaging due to their nanoscale arrangements, can help distinguish normal and pathological glomeruli for further proteomic analysis. The size of glomeruli in the human kidney is in the range of 110-280 μm (Samuel et al., J Anat. 2007), which perfectly fits the resolution capabilities of ProteomEx.

Reviewer #2 (Remarks to the Author):

The manuscript titled “Spatially Resolved Proteomics via Tissue Expansion” describes a method coupling tissue expansion with downstream spatial proteomics analysis. The dissection of spatially resolved tissue is performed using biopsy punches. The methods described here can certainly allow many laboratories with limited resources to explore spatially resolved proteomics as an option. The manuscript is well written and should be well received by the proteomics community.

We thank the Reviewer for the positive comments and constructive criticism.

However, a few important technical details are missing. Additionally, the authors tend to misleadingly over-represent their work as superior to all other methods. Their method sounds great and useful, and I anticipate a good adoption, nevertheless, in many aspects, it is not superior to some of the methodologies discussed in the paper. The authors should not only present the advantages of their method but also discuss its limitations.

In the course of the manuscript revision, we carried out additional experiments and data analysis, which helped us further validate and characterize the ProteomEx method as well as systematically compare it to other state-of-art methods for sample preparation. Correspondingly, we removed all statements that claimed the superiority of ProteomEx over other methods but rather presented it in the revised version as a more accessible alternative to existing methods for spatially resolved proteomics. We also expanded the Discussion sections describing the limitations and drawbacks of the current version of ProteomEx. For example, we downplayed the statements claiming that ProteomEx is a high-throughput method and that it has higher stability and reproducibility compared to PCT and in-solution digestion. In addition, we discussed potential limitations of ProteomEx workflow and spatial resolution. Please see detailed point-by-point responses below with references to the revised text and newly added results and Figures.

I hope that the following comment will help the authors to improve the quality and the impact of their manuscript.

Minor comments:

Line 63 the authors indicate that “data-independent acquisition (DIA) MS offers superior depth and throughput of proteomic analysis.” without precising over what. Looking at the methods presented, the sample preparation here will longer than the 12 hours of digestion and the runs are of an average duration of their runs is 2 pulseDIA of 50 min/sample which is longer than their DDA benchmark study. Every claim made in the paper stating that the method is faster than other methods is misleading.

We agree with Reviewer that this statement is not precise. We also agree that the presented method is not faster than other alternatives. Therefore, we rephrased this sentence by changing it to “data-independent acquisition (DIA) MS offers superior depth and reproducibility of proteomic analysis over data-dependent acquisition (DDA)”. In addition, in the revised **Figure 1** we added a detailed timeline of the ProteomEx workflow illustrating the duration of every step and hands-on time,

please see **Figure 1B**. All statements in the manuscript stating that the method is faster than other methods were changed to avoid claiming higher throughput of ProteomEx, the limitations of ProteomEx were also discussed in the revised main text. Please see P. 15-16 L. 338-342, and P. 16 L. 349-357.

An example of misleading statement (L68-71) is the following: “Several techniques based on these approaches can achieve up to ~50 μm of lateral resolution on thin (10’s of μm) tissue sections 20, however, they require special equipment not accessible to most labs. In addition, they are technically fragile and challenging in implementation, and have limited throughput and protein profiling depth.”

Only the NanoPOTS imaging method is cited here when the authors indicate that “several techniques” are available for such resolutions. They describe alternate methods as technically fragile without explaining why (this sounds more like an opinion than a reality). Similarly, the authors are indicating that LCM methods require special equipment not accessible to most labs, Mass spectrometry is another example of equipment that is not accessible to most lab, is expensive and require extensive expertise, yet their method utilize mass spectrometry. They state that alternate methods “have limited throughput” while the throughput of the example chosen (NanoPOTS) seems very similar to the ProteomEx one. As pointed by the authors in the evaluation of ProteomEx, the protein depth depends on the volume of tissue analyzed. By comparing the depth obtained from the smaller voxels 50 μm 2x10 μm to the much larger voxels of ProteomEx the authors mislead the reader to believe that ProteomEx is superior to other spatially-resolved methods altogether.

In the revised version of the manuscript, we removed all claims that ProteomEx has higher throughput or protein profiling depth compared to other methods for spatially resolved proteomics based on LCM and NanoPOTS. Instead, we presented a detailed comparison of protein identifications using different spatially resolved proteomics approaches providing all important experimental details including acquisition strategies (DDA and DIA), instrument types, software used, LC gradient lengths, as well as the dimension of the tissue samples and the number of identified peptides and proteins. Please see the newly added **Supplementary Table 4** and revised **Supplementary Figure 12**. We also explicitly stated that NanoPOTS has superior profiling depth for subnanoliter volumes of tissue compared to ProteomEx. We provided additional relevant citations to the sentence on P. 4 L. 68-69 and removed the sentence on P. 4 L. 70-71. Please also see revised text on P. 16 L.349-357.

Many other statements of this type across the paper would need to be tempered (examples : L332, 374, L229-334)

We revised these sentences to downplay their claims (see P. 15-16 L. 338-342; P. 18 L. 394-396).

Major comments:

The authors have not indicated the thickness of the sections they employed with their method. This is critical as they compare their method to the published one by Drelich et al. by simply comparing the peptide and protein counts to those obtained by this group. If the authors have used thicker brain sections (e.g., 16 μm) this could explain why their “peptide yield and protein identification” are better and may have nothing to do with the slight difference in employed technologies.

We thank Reviewer for pointing out this issue. In fact, comparison of the proExM-MS and ProteomEx methods was done using brain slices of the same thickness, i.e., 30 μm , and equal volumes of tissue were used. To be more specific, we added the corresponding statement to the manuscript “The proExM-MS and ProteomEx comparison was performed on the 30- μm thick brain slices”. Please see P. 21 L. 458-460.

L204 the other list the organs “brain, liver, and breast cancer” those are all “full” tissues. It would be great to evaluate potential delocalization for hollow organs such as Lungs for example as we can expect those to be more prone to deformation due to the empty space they contain.

To verify expansion distortion for hollow organs, we expanded mouse lung tissue using the ProteomEx protocol and performed isotropic analysis. The calculated root-mean-square length measurement over length scales of 1000-2000 μm was $\sim 10\%$, which is similar to that for liver tissue that can be considered as “full” tissue (**Figure 3A, B**). We added new results as **Supplementary Figure 8** and provide relevant description in the main text, please see P. 10 L. 208-209 and P. 10 L. 215-217.

Figure 2G-H and L192-195 the authors state “Albeit it was more peptides and proteins identified by PCT due to a larger sample injection amount, ProteomEx showed a higher degree of reproducibility in the processing of small sample volumes compared to PCT (Figure 2G, H). ProteomEx provides a new strategy for sub-nanoliter volume sample preparation for proteomic analysis.” When looking at the figure it seems clear that there is a single outlier for PCT that resulted in what the authors call “higher variability”. The punch cuts were realized on non-homogenous tissues and it is impossible to rule out that this specific cut was not done in a less dense region (e.g. overlapping with the exterior of the tissue) than all the other cuts. From the data presented without additional metrics, it is impossible to evaluate if PCT or ProteomeEx reproducibility is any different. The evaluation of the technical reproducibility has to be done on a homogenous tissue deposition rather than on an heterogenous tissue. Real metrics could then be discussed regarding the reproducibility of the method (CVs, sample-to-sample correlations, percentage of peptides/proteins identified in multiple runs, etc.).

We thank Reviewer for this great comment. To address this comment, 1) we performed additional analysis of the raw datasets represented in **Figure 2C, D**; 2) we carried out MS analysis of liver tissue microsamples processed with PrteomEx and compared it to macrosamples processed with PCT.

First, we reanalyzed our DDA files from the four methods (In-solution, PCT, proExM-MS, ProteomEx) using the MSFragger software in the label-free quantification (LFQ) mode. We presented the Pearson correlation between each pair of samples as a heatmap and coefficient of variation (CV) values were visualized using violin plots (see the Figure below). The gel punches from the adjacent microregions sampled by ProteomEx (5.9 nL) were also added for comparison.

The Pearson correlations of the replicates from the same methods were mostly greater than 0.90, except for one sample prepared by the in-solution strategy, which showed a higher deviation than the other methods (**Supplementary Figure 6**). The samples processed with ProteomEx achieved relatively high Pearson correlation values of about 0.94-0.96 in the quarter of the slice (250 nL) group. The correlation values of the microregion samples (5.9 nL) were in the range of 0.89-0.92, which was lower than the values of the quarter of the slice (250 nL) most likely due to the heterogeneity of the mouse brain tissue. Overall, these values were higher or comparable to other three methods used for benchmarking, *i.e.*, 0.89-0.95 for proExM-MS, 0.90-0.95 for PCT, and 0.81-0.96 for in-solution. The median CV of protein intensity (red point in the violin plots) is the lowest for the quarter of slice samples prepared with ProteomEx, which indicated the highest reproducibility of this method (see below).

Supplementary Figure 6. Reproducibility and stability comparison for the selected sample preparation methods. (A) Heatmap of Pearson correlations for protein quantification for each pair of samples from the four sample preparation methods analyzed using the MSFragger software ($n=4, 4, 7, 4, 3$ technical replicates from one, one, two, one, and one brain slices for in-solution digestion, PCT, proExM-MS, ProteomEx, and ProteomEx (5.9 nL sample), respectively; the MS raw files corresponding to Figure 2C were used for analysis). The color bar indicates the values of Pearson correlations. (B) Coefficient of variation of quantified protein abundance from the four methods.

For further evaluation of the technical reproducibility of ProteomEx for microsamples we used mouse liver tissue, which is more homogenous than brain tissue. Tissue samples acquired from the same mice were processed according to the established protocols and analyzed using the PulseDIA method. With ProteomEx we processed four different tissue volumes from 2.75 to 17.19 nL while samples processed with PCT were about two orders of magnitude larger from 200 to 1,000 nL. For quantitative comparison, we calculated the overlapped protein ratios, Pearson correlation for protein quantification between each pair of samples and coefficient of variation (CV) values.

In the case of ProteomEx, the overlapped identified protein ratios for 3 independent runs were 62.9%, 66.7%, 66.7% and 70.3% for the tissue volumes of 2.75 nL, 6.19 nL, 11.00 nL and 17.19 nL, respectively. The larger volume of tissue samples are processed the more identified proteins overlap. Correspondingly, the Pearson correlations of the replicates from the same sample size were greater than 0.90 with smaller samples (2.75 nL) having lower values (0.90-0.93) and larger samples (17.19 nL) having higher values (0.94-0.96). The samples processed with PCT achieved slightly higher Pearson correlation values in the range of 0.89-0.98 with no dependence on sample size. Furthermore, the median CVs of protein intensity (red point in the violin plots shown below) were lower for samples processed with PCT. These results indicated that ProteomEx has similar or slightly lower reproducibility for microsamples compared to macrosamples processed with PCT. However, at this point, we cannot rule out if the lower reproducibility for the smallest used samples for ProteomEx was due to the heterogeneity of the liver tissue or technical reproducibility. Nevertheless, we agree with the Reviewer that the statement regarding ProteomEx reproducibility in the original manuscript might be misleading therefore we removed it and added new data and conclusions stating that ProteomEx has similar or slightly lower reproducibility for microsamples and macrosamples when compared to other methods for macrosample processing. Please see the revised main text P. 9 L. 184-191 and the newly added **Supplementary Figure 6 and 7**.

Supplementary Figure 7. Reproducibility and stability of ProteomEx for microsamples. (A,B) Heatmap of Pearson correlations for protein quantification for each sample pair from (A) ProteomEx and (B) PCT analyzed by PulseDIA (the overlapped identified protein ratios in 3 independent runs were 62.9%, 66.7%, 66.7%, and 70.3% for the tissue volumes of 2.75 nL, 6.19 nL, 11.00 nL, and 17.19 nL, respectively). The color bar indicates the values of Pearson correlations (n=3 adjacent slices from 1 mouse for each method). (C) Coefficient of variation of quantified protein abundance from the two methods (n=3 slices from 1 mouse for each method).

The authors admit (L320) that “manual handling of small transparent gel samples, which were hard to see by the naked eye” limit the resolution to which samples can be obtained. The size of the tissue disks obtained is constrained by the availability of punch cutters. I would be very

surprised to see how the method can be implemented to cut precisely with the naked eye 50 single glomerulus of 150 to 300 μm of diameter to perform single glomerulus proteomics. LCM technologies are compatible with immunofluorescence for the identification of specific cells/tissues organization and enable a much more precise dissection of tissues. The ProteomeEx method can only be used to perform spatially resolved sections on relatively homogenous tissue regions. This major limitation among other that can be noted should be discussed in the manuscript.

We agree with Reviewer that ProteomeEx has several limitations compared to LCM in terms of achieved spatial resolution and precise targeting of less than $\sim 100 \mu\text{m}$ in size tissue regions based on morphological features such as, for example, glomeruli. Correspondingly, we added this point to the Discussion section of the revised manuscript. Please see P. 15-16 L. 338-342.

Reviewer #3 (Remarks to the Author):

Summary:

This work combined the ability of hydrogel expansion with mass spectrometry based proteomics. The authors showed a modified gel recipe with improved expansion factors, Coomassie staining of the expanded hydrogels and microdissection/ microdigestion protocols. As mentioned and discussed by the authors, this paper conceptually overlaps with Drelich's work [1] from last year, in which the expansion combined with mass spectrometry based proteomics was also shown. The authors mentioned in the manuscript that the two researches differ mainly in 1) Expansion protocol, 2) higher peptide yield and 3) accurate microdissection with the colorimetric staining, and performed side-by-side comparison.

We thank the Reviewers for appreciating our work and for providing valuable comments to improve the manuscript.

There are still some general questions that might need to be clarified:

1. The expansion factor (EF) mentioned in the paper is from 5.5 to 8 with tissue embedding and the test of new gel composition is with final expansion factor of an empty gel from 5.3 to 8.4. And in the text, there was one $EF = 8$ image shown in S4, the rest analyzed data or images was however with only lower magnification. This is relatively big variance compared to the other ExM techniques. Is that due to monomer quality difference between batches or from some other issue, such as incomplete homogenization? And if most of the experiments are not characterized with the 8x protocol (e.g. peptide retention, distortion analysis), it would be appropriate to change the discussion/calculation in the maintext with the more reproducible expansion factor.

We have carefully measured and documented linear expansion factors for all samples reported in the manuscript. The expansion factors for the same batch of samples were very consistent. For example, the brain samples reported in **Figure 4** had highly reproducible expansion factors of 5.9 ± 0.2 and 5.8 ± 0.2 for WT and AD mice, respectively. In the revised manuscript, we indicated expansion factors for all samples reported in the main text figures, please see revised figure legends

for **Figure 1, 2, 3, 4**. Indeed, expansion factor of the tissue-hydrogel composite was lower than pure hydrogel reported in **Supplementary Table 1**, and on average for all experiments reported in this study it was 6.1-fold. Therefore, we revised the main text to indicate the average expansion factor for most of the reported samples, please see P. 4 L. 77-78 of the revised manuscript.

2. The term “reversible protein anchoring to polymer” is used to describe N-Succinimidyl acrylate, but I didn't find why it is “reversible”. Although NSA was used in ExM for the first time, a very similar chemical Methacrylic acid NHS ester was published since 2016 and used in tissues ExM in multiply labs ([2],[3]). As my understanding this chemical is even cheaper, also relatively stable. Did the authors already compared this anchoring molecule MA-NHS with the NSA?

The NSA anchor should primarily modify primary amine groups of the amino acid side chains, such as Lys (K), Asn (N), Gln (Q), and Arg (R), which would serve as attachment points to the polymer mesh. To investigate detaching peptides from the polymer network, we compared the ratio of peptides containing K, N, Q, and R amino acids extracted from the expanded tissue with that for samples prepared using in-solution digestion and PCT where chemical modification of amino acids was not used (for this we used data presented in **Figure 2C, D**). We observed that the ratios of the peptides containing N, Q, and R were almost identical for all used methods (see the plot below). However, the ratios of the peptides containing K were slightly lower for proExM-MS (54.04%) and ProteomEx (56.54%) methods than that for in-solution digestion (63.47%) and PCT (62.51%). For reference, we also compared the ratio of lysine-containing peptides extracted from the expanded gels homogenized either with SDS-containing buffer or with TFE (data from **Supplementary Figure 2**). We revealed that ratio for SDS-based homogenization corresponding to the ProteomEx protocol was 57.3%, which is similar to that for samples in **Figure 2**. However, in case of TFE treated samples the ratio was almost twice lower about 27.5%, which might indicate incomplete retrieval of peptides. These results indicated that the protein anchoring in the ProteomEx protocol is reversible and provides retrieval efficiency of the peptides that covalently anchored to the polymer network comparable with the common sample preparation methods, such in-solution digestion and PCT. To support our statement regarding reversible protein anchoring to polymer we revised the manuscript to add data for peptide retrieval to different methods. We have not compared MA-NHS with NSA for proteomic analysis, but we believe these chemical anchors should have similar performance.

Supplementary Figure 5. Post-translationally and chemically modified peptide analysis for tissue samples processed with four selected methods. (A) Fraction of identified peptides containing selected amino acids. The values for each column represent the mean, error bars represent standard deviation. (B) Anchor-modified peptide analysis. Y-axis indicates the anchor-modified peptide fraction (%). The values on the top of the sections are mass shift delta, 54.0474 for ProteomEx corresponding to the modification with NSA anchor, 114.1656, and 168.2130 for proExM-MS corresponding to the modification AcX anchor. The values for each column represent the mean, error bars represent standard deviation. *P*-values are estimated by Welch's t-test (**P*<0.05, ***P*<0.01, ****P*<0.001, pairs without indicated *P*-value are statistically non-significant, *i.e.*,

P>0.05). (C) The Upset plot with the numbers of overlapped post-translational modification of peptides for the four methods. (D) The heatmap illustrating the percentage of different modifications of peptides. Each row represents a type of peptide modification. The rows and columns are clustered by the hierarchical method.

3. The distortion is not as important in this study compared to optical based ExM. But the distortion is still relatively larger (10%) compared to many of the published expansion protocol for tissues (5% or less). This will still significant limit the new recipe to be broadly used in microscopy related studies. Did the authors maybe also perform some distortion analysis during the gelation composition selection? Or can this gel composition be further improved for a lower distortion over long distance?

We agree with Reviewer that isotropic expansion is crucial for microscopy-related studies when super-resolution imaging is required. Therefore, isotropic expansion is usually characterized on microscale, *i.e.*, 10-100 μm . Indeed, most of the ExM protocols have a distortion rate less than 5%. For example, in proExM r.m.s. length measurement error for pancreas on 100 μm scale was $\sim 1.5\%$, for spleen on 25 μm scale was 2.8%, for lungs on 40 μm scale was 2.0% (Tillberg et al., 2016). For tetra-gel based proExM distortion was about 2.5% over 20 μm scale when measured in cell culture (Gao et al., 2021). However, for the ExM protocols characterized by larger expansion factors, such iExM (Chang et al., 2017) and TREx (Damstra et al., 2022), distortion was almost twice higher, about 4%. The distortion of expansion for ProteomEx was measured for 6-8-fold expanded samples over a large length scale, about 1500 μm , to match the scale of microdissection, which is significantly larger than that used for characterization of the previously reported ExM protocols for super-resolution imaging (**Figure 3**). We did not perform distortion analysis during the gelation composition selection, although we think is it possible to further improve the ProteomEx gel composition for a lower distortion over long distances specifically if combination with super-resolution imaging will be required. Nevertheless, we think this is an important point, therefore we revised the Discussion section of the manuscript to provide comparison of isotropic expansion of ProteomEx with other ExM protocols for super-resolution imaging. Please P. 15-16 L. 338-342.

4. It was mentioned in the SI that reducing the concentration of crosslinker will result in more fragile gels that may not show uniform expansion. There was a new study [4] that proposed the modification with low crosslinker concentration, better handling and near 10x expansion with the conventional chemicals. The author might need to add the paper as a reference.

We thank the Reviewer for the helpful comment. We cited the recent work by Damstra et al. in the corresponding context. Please see revised **Supplementary Note 1** P. 2 L. 57-60.

5. The authors claimed the screening of 400 hydrogels with the criteria expansion factor and mechanical stability. Current ExM are mostly based on sodium acrylate, but it was completely filtered out. If sodium methacrylate is significant better, it would be better if the example of being mechanically stable can be shown as video or figures.

To perform quantitative comparison of the hydrogel mechanical stability, we used the compression testing machine (CTM6050, Xie Qiang Instrument Manufacturing Co, China) equipped with a S9M/1kg force sensor (HBM, Germany). For comparison with the ProteomEx hydrogel, we used two sodium acrylate containing hydrogels. First, we used the ExM hydrogel, which is the most widely used hydrogel for tissue expansion (Tillberg et al., 2016). As second hydrogel we used ExM hydrogel with addition of N,N-dimethylacrylamide, which, as was discovered during hydrogel screening, allowed for higher expansion factor (see **Supplementary Note 1**). The expanded hydrogels were slowly compressed to measure the strain and stress at breaking point. It was revealed that the ExM hydrogel breaks at 49% of compression, while DMAA containing and ProteomEx hydrogel remain stable until 81 and 77% of compression, respectively. The stress achieved at breaking point was similar for all three hydrogels on average ~19 kPa. However, it should be noted that expansion factor of ProteomEx hydrogel was higher (6.25 linear expansion factor) than other hydrogels (4.0 for ExM and 4.75 for DMAA containing ExM hydrogel). Overall, ProteomEx hydrogel exhibits significantly higher expansion factor while similar or slightly higher mechanical stability than sodium acrylate based hydrogels and therefore was selected for tissue expansion. Please see newly added **Supplementary Figure 1** with stress-strain curves and corresponding descriptions in Method section of the main text (P. 20 L. 436-438) and **Supplementary Note 1** (P. 3-4 L. 110-118).

Some small points:

1. It was mentioned in the SI note 1, “We ultimately selected the two hydrogel recipes consisting of SMA:DMAA:PAE in molar ratio 1:4:0.0008 and SMA:DMAA:TPT in molar ratio 1:4:0.0005, which were characterized by linear expansion factors of 8.2 and 8.4, respectively.” But in the SI table 1, only 1:4:0.0008 was labelled as a ProteomEx.

Indeed, based on the pure hydrogel screening we selected two compositions, which were further tested with biological samples for efficiency of peptide extraction and protein identification using MS analysis. Based on MS data we selected only one hydrogel, namely SMA:DMAA:PAE, for further characterization and benchmarking. The results of selection of the SMA:DMAA:PAE hydrogel for MS analysis are presented in **Supplementary Table 2** and **Supplementary Figure 2** and described in **Supplementary Note 1**. To be more precise in description, we revised **Supplementary Note 1** to remove the word “ultimately”, which might be misleading, and indicated the hydrogel composition selected for all experiments presented in the main text figures. Please see P. 3 L. 105-106 and 110-114 of **Supplementary Materials**.

2. The coomassie staining part was quite interesting, which facile the precise cutting, but also requires more handling on the gel, staining and destaining. I wonder if the destaining step or treatment with methanol etc would influence the stiffness of the hydrogel.

We did not observe any changes in the mechanical stability of the hydrogels after washing with methanol. We added this information into the Methods section of the revised manuscript.

3. The part for in-gel digestion is also very nice. LysC and trypsin was previously used for homogenization, here the authors cleverly used them for peptide extraction.

We thank the Reviewer for the kind comment. Indeed, we intentionally did not use proteolytic digesting for homogenization to retain proteins in the expanded state.

4. The proteome analysis seems to be based on the structures identified by naked eye with coomassie staining. By how much would the local distortion or heterogeneous expansion in hydrogel influence the proteome analysis result?

Based on the analysis of tissue distortion during expansion reported in **Figure 3A**, the length measurement errors over length scales used for spatially resolved proteomics of tissues were no more than 6-10%. Therefore, we suggested that the local distortion and heterogeneous expansion in hydrogel should not influence the proteome analysis result by more than 10%. We added this point to the Discussion section of the revised manuscript. Please see P. 15 L. 322-326.

[1] Drelich et al. *Acs anal. chem*, 2021

[2] Fan et al., chapter 7 in *Expansion Microscopy for Cell Biology* (2021)

[3] Chozinski et al. *Nat. Methods*, 2016

[4] Damstra et al. *Elife*, 2022

REVIEWERS' COMMENTS

Reviewer #1 (Remarks to the Author):

I thank the authors for addressing my concerns and congratulate them on their work.

Reviewer #2 (Remarks to the Author):

I am delighted that the remarks made during the review process helped the author to make the manuscript stronger. I feel that the manuscript was strengthened and I believe it should be accepted in its current state. The manuscript should be well received by the spatial proteomics community.

This is only a suggestion to the authors but I believe that a very detailed protocol for the different steps of the process on an open protocol platform (such as protocol.io) would help the adoption of the technique by many other laboratories and proteomics teams.

Reviewer #3 (Remarks to the Author):

The re-submitted manuscript addresses most of my concerns. The authors established an expansion-based LC-MS termed 'ProteomEx' and showed side-by-side comparison to the other techniques. It has been successfully established with the existing and the newly added figures that the new hydrogel recipe could enable tissue expansion with higher expansion factor (6x compare to 3-4x), better mechanical stability and tolerable distortion over longer distance (10% compare to 4%) compare to the original ExM protocol (Tillberg et al. 2015). And the application in LC-MS based proteomics also succeeded with higher peptide yields and spatio resolution. The methods described in manuscript would be beneficial for readers from different fields. I therefore recommend the publication on Nature Communication.

There are some textual concerns left, which are not essential, but I think they could improve the quality of the paper.

The expansion factor of 8 is not further established or explained. This was the expansion factor used to claim the resolution of the ProteomEx in abstract and throughout the whole article. But in the detailed figures and results, it was only shown in one gel composition test (Table 1), one expanded brain tissue (Figure S10) and the corresponding analysis (Fig 3E). The rest of the data, as the authors also mentioned in the reply, was with very consistent expansion factors around 6.1, including the distortion analysis, parallel comparisons and validation in different tissues. In the newly added mechanical stability test, the empty hydrogel with the same composition as in Table 1 was also only expanded to 6.25 rather than 8.1. Thus personally I cannot see the point of using a single data set with 8x expansion factor to summarize the whole study. I would recommend to change most of the phrases in the abstract, discussion and summary containing '8x', '125 μ m linear resolution', '512-fold expansion in volume' or '0.37 nL' to the real reproducible expansion factor '6.1x' and the corresponding resolution/volume. This won't weaken the significance of the paper, but rather highlight the reproducibility and accuracy of ProteomEx.

Other points:

In the main-text, page 23, line 495. Perhaps the authors meant by 'not changing?'

It would be helpful to have both volume of hydrogel and volume of the calculated dimension pre-expansion.

We thank all Reviewers and Editorial team for appreciating our work and for providing helpful comments to improve the manuscript. We have addressed the remaining Reviewers' comments. Below we provide point-by-point response, the corresponding edited text in the manuscript is highlighted with blue.

Reviewer #1 (Remarks to the Author):

I thank the authors for addressing my concerns and congratulate them on their work.
We thank Reviewer for appreciating our work and providing valuable feedback.

Reviewer #2 (Remarks to the Author):

I am delighted that the remarks made during the review process helped the author to make the manuscript stronger. I feel that the manuscript was strengthened and I believe it should be accepted in its current state. The manuscript should be well received by the special proteomics community.
We thank Reviewer for valuable comments and positive feedback on our technology.

This is only a suggestion to the authors but I believe that very detailed protocol for the different steps of the process on an open protocol platform (such as protocol.io) would help the adoption of the technique by many other laboratories and proteomics team.

This is a great suggestion. We are working on the detailed step-by-step protocol describing the ProteomEx workflow. We will share it online as soon as possible.

Reviewer #3 (Remarks to the Author):

The re-submitted manuscript addresses most of my concerns. The authors established an expansion-based LC-MS termed 'ProteomEx' and showed side-by-side comparison to the other techniques. It has been successfully established with the existing and the newly added figures that the new hydrogel recipe could enable tissue expansion with higher expansion factor (6x compare to 3-4x), better mechanical stability and tolerable distortion over longer distance (10% compare to 4%) compare to the original ExM protocol (Tillberg et al. 2015). And the application in LC-MS based proteomics also succeeded with higher peptide yields and spatio resolution. The methods described in manuscript would be beneficial for readers from different fields. I therefore recommend the publication on Nature Communication.

We appreciate the positive feedback and thank Reviewer for the thoughtful suggestions, which we address in the revised version of the manuscript. Please see below our detailed response.

There are some textual concerns left, which are not essential, but I think they could improve the quality of the paper.

The expansion factor of 8 is not further established or explained. This was the expansion factor used to claim the resolution of the ProteomEx in abstract and throughout the whole article. But in

the detailed figures and results, it was only shown in one gel composition test (Table 1), one expanded brain tissue (Figure S10) and the corresponding analysis (Fig 3E). The rest of the data, as the authors also mentioned in the reply, was with very consistent expansion factors around 6.1, including the distortion analysis, parallel comparisons and validation in different tissues. In the newly added mechanical stability test, the empty hydrogel with the same composition as in Table 1 was also only expanded to 6.25 rather than 8.1. Thus personally I cannot see the point of using a single data set with 8x expansion factor to summarize the whole study. I would recommend to change most of the phrases in the abstract, discussion and summary containing '8x', '125 μm linear resolution', '512-fold expansion in volume' or '0.37 nL' to the real reproducible expansion factor '6.1x' and the corresponding resolution/volume. This won't weaken the significance of the paper, but rather highlight the reproducibly and accuracy of ProteomEx.

We agree with the Reviewer's suggestion. We have revised Abstract, Discussion, and Summary to indicate the real reproducible expansion factor '6.1x' and the corresponding later resolution and volume. Please see revised text P. 2 L. 38, P. 15 L. 331-332, and P 17 L. 372.

Other points:

In the main-text, page23, line 495. Perhaps the authors meant by 'not changing?'

We thank Reviewer for pointing out this typo, we fixed this mistake.

It would be helpful to have both volume of hydrogel and volume of the calculated dimension pre-expansion.

We have added volume of the expanded tissue excised from the tissue-hydrogel composite and corresponding calculated volume of tissue before expansion. Please see revised text P. 10 L. 199-202.